# Terpenoid and carbonyl emissions from Norway spruce in Finland during the growing season

Hannele Hakola[1], Virpi Tarvainen[1], Arnaud P. Praplan[1], Kerneels Jaars[2], Marja Hemmilä[1], Markku Kulmala[3], Jaana Bäck[4], Heidi Hellén[1]

[1]Finnish Meteorological Institute, Atmospheric Composition Unit, P.O. Box 503, 00101 Helsinki, Finland
[2]Unit for Environmental Sciences and Management, North-West University, Potchefstroom, South Africa
[3]Department of Physics, P.O. Box 64, 00014 University of Helsinki, Finland
[4]Department of Forest Ecology, P.O. Box 27, 00014 University of Helsinki, Finland

*Correspondence to*: Hannele Hakola (hannele.hakola@fmi.fi)

**Abstract.** We present spring and summer volatile organic compound (VOC) emission rate measurements from Norway spruce (*Picea abies* L. Karst) growing in a boreal forest in southern Finland. The measurements were conducted using in situ gas-chromatograph with 1to 2-hour time resolution to reveal quantitative and qualitative short-term and seasonal variability of the emissions. The measurements cover altogether 14 weeks in years 2011, 2014 and 2015. Monoterpene (MT) and sesquiterpene (SQT) emission rates were measured all the time, but isoprene only in 2014 and 2015 and acetone and $C_4$-$C_{10}$ aldehydes only in 2015. The emission rates of all the compounds were low in spring, but MT, acetone and $C_4$-$C_{10}$ aldehydes emission rates increased as summer proceeded, reaching maximum emission rates in July. Late summer mean values (late July and August) were 29, 17 and 33 ng g(dw)$^{-1}$ h$^{-1}$ for MTs, acetone and aldehydes respectively. SQT emission rates increased during the summer and highest emissions were measured late summer (late summer mean value 84 ng g(dw)$^{-1}$ h$^{-1}$) concomitant with highest linalool emissions most likely due to stress effects. The between-tree variability of emission pattern was studied by measuring seven different trees during the same afternoon using adsorbent tubes. Especially the contributions of limonene, terpinolene and camphene were found to vary between trees, whereas proportions of α-pinene (25±5 %) and β-pinene (7±3 %) were more stable. Our results show that it is important to measure emissions on canopy level due to irregular emission pattern, but reliable SQT emission data can be measured only from enclosures. SQT emissions contributed more than 90 % of the ozone reactivity most of the time, and about 70 % of the OH reactivity during late summer. The contribution of aldehydes to OH reactivity was comparable to the one of MT during late summer, 10 %-30 % most of the time.

## 1 Introduction

The boreal forest is the largest terrestrial biome, forming an almost continuous belt around the northern hemisphere. The boreal forest zone is characterized by a short growing season and a limited number of tree species. The most common tree species are Scots pine, Norway spruce and silver and downy birch and they produce and emit vast amounts of biogenic volatile organic compounds (VOCs) (Bourtsoukidis et al., 2014a, b; Bäck et al., 2012; Cojocariu

et al., 2004; Grabmer et al., 2006; Hakola et al., 2001, 2006; Tarvainen et al., 2005; Yassaa et al., 2012). The
compounds are mainly isoprene, monoterpenes (MT), sesquiterpenes (SQT) and oxygenated volatile organic
compounds (OVOCs) (Tarvainen et al., 2007). There is a variety of factors controlling these emission, both biotic
(Pinto-Zevallos et al., 2013; Joutsensaari et al., 2015) and abiotic stress (Vickers et a., 2009; Bourtsoukidis et al.,
2012; Bourtsoukidis et al., 2014c) factors can initiate or alter VOC emissions. Abiotic stress factors have been
reviewed by Loreto and Schnizler (2010). Terpenes for example relieve oxidative and thermal stresses of trees. Many
stress factors can also interact and cause additive effects (Niinemets, 2010; Holopainen and Gershenzon, 2010).
Biotic stresses such as acarid species infestation have been shown to initiate farnesene and linalool emissions in
spruce seedlings (Kännaste et al., 2008).  Emission potentials and composition varies a lot between different tree
species (Guenther et al. 2012). However, there is also a lot of variation in the emissions of different individuals of the
same tree species. Bäck et al. (2012) showed that Scots pine trees of the same age, growing in the same environment,
emit very different monoterpene selections. These so called different chemotypes cause uncertainties in emission
modelling.
In the atmosphere VOCs are oxidized, which affects the tropospheric ozone formation (Chameides et al., 1992) and
contribute to the lifetime of methane by consuming hydroxyl radicals. In addition reaction products of VOCs also
participate in the formation and growth of new particles (Tunved et al., 2006). In smog chamber studies secondary
organic aerosol (SOA) yields for different hydrocarbons and even for different MTs have been found to vary
considerably (Griffin et al., 1999). Jaoui et al. (2013) studied SOA formation from SQT and found that the high
reactivity of SQT produced generally high conversion into SOA products. Furthermore, they found that the yields were
dependent on the oxidant used and were highest for nitrate radical ($NO_3$) reactions. Of the SQT acidic products, only
β-caryophyllinic acid has been observed in ambient samples (Jaoui et al., 2013; Vestenius et al., 2014). Due to their
high reactivity, SQT are not usually found in ambient air. Hakola et al. (2012) detected longifolene and isolongifolene
in boreal forest air during late summer. Hence, the best way to evaluate the atmospheric impact of SQTs is to measure
them from emissions.
In addition to isoprene and MTs and SQTs, plants emit also large amounts of oxygenated compounds i.e. alcohols,
carbonyl compounds and organic acids (Koppmann and Wildt, 2007). OVOCs containing six carbon atoms ($C_6$) are
emitted directly by plants often as a result of physical damage (Fall et al., 1999; Hakola et al., 2001). Saturated
aldehydes (hexanal, heptanal, octanal, nonanal, and decanal) have also been found in direct emissions of plants (Wildt
et al., 2003) as well as methanol, acetone and acetaldehyde (Bourtsoukidis et al. 2014b).
In the present study we conducted on-line gas-chromatographic measurements of emissions of MTs and SQTs as well
as $C_4$-$C_{10}$ saturated aliphatic carbonyls from Norway spruce (*Picea abies* L. Karst) branches. Although Norway spruce
is one of the main forest tree species in Central and Northern Europe, there are relatively limited amount of data on its
emissions (Hakola et al., 2003; Grabmer et al., 2006; Bourtsoukidis et al., 2014a and b, Yassaa et al. 2012). Rinne et
al. (2009) identified knowledge gaps concerning VOC emissions from boreal environment and concluded that there is
a lack of knowledge in non-terpenoid emissions from most of the boreal tree species.  They also pointed out that
chemotypic variations are not well enough understood to be taken into account in emission modelling. To fill this
knowledge gap we conducted BVOC emission measurements from Norway spruce. On-line gas-chromatograph mass
spectrometer (GC-MS) was chosen because in addition to detection of individual MTs it allows sensitive detection of
SQTs, which is often difficult to perform under field conditions. The on-line measurements were considered essential
for evaluating the factors affecting emission rates, for example their temperature and light dependence. Our campaigns
cover periods of years 2011, 2014 and 2015 during spring and summer, altogether about 14 weeks. In 2015 also
carbonyl compounds were added to the measurement scheme, since there is no earlier data of their emissions.
**2 Methods**
**2.1 VOC measurements**
The measurements were conducted at the SMEAR II station (Station for Measuring Forest Ecosystem-Atmosphere
Relations, $61^0$51'N, $24^0$18'E, 181 a.s.l) in Hyytiälä, southern Finland (Hari and Kulmala 2005) in 2011, 2014, and
2015. The measurements took place in spring/early summer 2011 (two weeks in April, five days in May and three days
in June), spring/summer 2014 (one week in May, two weeks in June and one week in July), and summer 2015 (one
week in June and two weeks in August) and they were conducted using an in situ gas-chromatograph.
Two different trees were measured; tree 1 in 2011 and tree2 in 2014 and 2015. The selected trees were growing in a
managed mixed conifer forest (average tree age ca 50 years), and located about 5 meters from the measurement
container. The height of the tree 1 in 2011 was about 10 meters (age about 40 years). The measured branch was a
fully sunlit, healthy lower canopy branch pointing towards a small opening at about 2 meters height. In 2014 and
2015 a younger tree (tree 2, ca. 1 m tall, age ca 15 years) about 5 meters away from the tree used in 2011 was
selected for the study. The branches were placed in a Teflon enclosure and the emission rates were measured using a
dynamic flow through technique. The setup is shown in Fig. 1. The volume of the cylinder shape transparent Teflon
enclosure was approximately 8 litres and it was equipped with inlet and outlet ports and a thermistor (Philips KTY
80/110, Royal Philips Electronics, Amsterdam, Netherlands) covered with Teflon tubing inside the enclosure. The
photosynthetically active photon flux density (PPFD) was measured just above the enclosure by quantum sensor (LI-
190SZ, LI-COR Biosciences,Lincoln, USA).
The flow through the enclosure was kept at about 3-5 litres min$^{-1}$. Ozone was removed from the incoming air using
manganese oxide ($MnO_2$) coated copper nets. The emission rates were measured using the on-line GC-MS. From the
enclosure outlet port air was directed through the 6 m long fluorinated ethylene propylene (FEP) inlet line (i.d. 1/8
inch) to the GC-MS with the flow of ~0.8 L/min. Subsamples were taken from this main flow with the flow of 40-60
ml/min directly into the cold trap of a thermal desorption unit (Perkin Elmer ATD-400) packed with Tenax TA in 2011
and Tenax TA/Carbopack-B in 2014 and 2015. The trap material was changed since isoprene was found not to be
retained fully in the cold trap in 2011. The trap was kept at $20^0$C during sampling to prevent water vapour present in
the air from accumulating into the trap. The thermal desorption instrument was connected to a gas chromatograph (HP
5890) with DB-1 column (60 m, i.d. 0.25 mm, f.t. 0.25 µm) and a mass selective detector (HP 5972). One 20-minutes
sample was collected every other hour. The system was calibrated using liquid standards in methanol injected on Tenax
TA-Carbopack B adsorbent tubes. The detection limit was below 1 pptv for every MT- and SQT.
The following compounds were included in the calibration solutions: 2-methyl-3-buten-2-ol (MBO) (Fluka), camphene
(Aldrich), 3-carene (Aldrich), p-cymene (Sigma-Aldrich), 1,8-cineol (Aldrich), limonene (Fluka), linalool (Aldrich),
myrcene (Aldrich), α-pinene (Sigma-Aldrich), β-pinene (Fluka), terpinolene (Fluka), bornylacetate (Aldrich),
longicyclene (Aldrich), isolongifolene (Aldrich), β-caryophyllene (Sigma), aromadendrene (Sigma-Aldrich), α-
humulene (Aldrich), β-farnesene (Chroma Dex). Isoprene was calibrated using gaseous standard from National
Physical Laboratory (NPL). We had no standard for sabinene and therefore it was quantified using the calibration curve
of β-pinene, because both species elute close each other and their mass spectra are similar. Therefore the results for
sabinene are only semi-quantitative, but it enables the observations of diurnal and seasonal changes. Compared to off-
line adsorbent methods this in situ GC-MS had clearly lower background for carbonyl compounds and in 2015 we
were able to measure also acetone/propanal and $C_4$-$C_{10}$ aldehyde emission rates. The aldehydes included in the
calibration solutions were: butanal (Fluka), pentanal (Fluka), hexanal (Aldrich), heptanal (Aldrich), octanal (Aldrich),
nonanal (Aldrich) and decanal (Fluka). Unfortunately, acetone co-eluted with propanal and the calibration was not
linear due to high acetone background in adsorbent tubes used for calibrations.
**2.2 Calculation of emission rates**
The emission rate is determined as the mass of compound per needle dry weight and per time according to

$E = \frac{(C_2 - C_1)F}{m}$ (1)

Here $C_2$ is the concentration in the outgoing air, $C_1$ is the concentration in the incoming air, and $F$ is the flow rate
into the enclosure. The dry weight of the biomass ($m$) was determined by drying the needles and shoot from the
enclosure at 75 ℃ for 24 hours after the last sampling date. We also measured needle leaf areas and the specific leaf
area SLA is 136 $m^2$ $g^{-1}$.
**2.3 Emission potentials**
A strong dependence of biogenic VOC emissions on temperature has been seen in all emission studies of isoprene,
MTs, and SQTs (e.g. Kesselmeier and Staudt 1999; Ciccioli et al. 1999; Hansen and Seufert 2003; Tarvainen et al.
2005; Hakola et al. 2006). The temperature dependent pool emission rate is usually parameterized using a log-linear
formulation


$$E(T) = E_S \exp(\beta(T - T_S)) \tag{2}$$

where $E(T)$ is the emission rate (µg g$^{-1}$ h$^{-1}$) at leaf temperature $T$ and $\beta$ is the slope $\frac{d\,lnE}{dT}$ (Guenther et al. 1993). $E_S$ is
the emission rate at standard temperature $T_S$ (usually set at 30 °C). The emission rate at standard temperature is also
called the emission potential of the plant species, and while it is sometimes held to be a constant it may show variability
related to e.g. season or the plant developmental stage (e.g. Hakola et al. 1998, 2001, 2003, 2006; Tarvainen et al.
2005, Aalto et al 2014).
Besides the temperature-dependent nature of the biogenic emissions, light dependence has been discovered already in
early studies of plant emissions (e.g. the review of biogenic isoprene emission by Sanadze 2004 and e.g. Ghirardo et
al 2010). The effect of light on the emission potentials is based on the assumption that the emissions follow similar
pattern of saturating light response which is observed for photosynthesis, and the formulation of the temperature effect
is adopted from simulations of the temperature response of enzymatic activity. The algorithm formulation is given e.g.
in Guenther et al. 1993 and Guenther 1997.
In this work we have carried out nonlinear regression analysis with two fitted parameters, arriving at individual
standard emission rates and slope values for the modelled MTs and SQTs compounds during each model period. The
compounds analysed with the temperature dependent pool emission rate were the most copiously emitted MTs and
SQTs, other MTs, other SQTs, acetone and sum of aldehydes.  The light and temperature controlled instant emission
rates were obtained for isoprene. An alternative modelling approach was tested using a hybrid emission algorithm,
which has both the temperature-dependent pool emission and the light and temperature controlled instant emission
terms.



**2.4 Chemotype measurements**
In order to estimate the between-tree variability of the emissions, we conducted a study in 2014, where we made
qualitative monoterpene analysis from six different spruces (trees 3-8) growing in a same area not farther than about
10 metres from each other. All the trees were about 1 m high and naturally regenerated from local seeds. A branch was
enclosed in a Teflon bag and after waiting for 5 minutes we collected a 5 minute sample on a Tenax TA/Carbopack-B
tube and analysed later in a laboratory using Perkin-Elmer thermodesorption instrument (Turbomatrix 650) connected
to Perkin-Elmer gas-chromatograph (Clarus 600) mass spectrometer (Clarus 600T) with DB-5 column. The samples
were taken during one afternoon on 24 June 2014.

**2.4 Calculating the reactivity of the emissions**

We calculated the total reactivity of the emissions ($TCRE_x$) by combining the emission rates ($E_i$) with reaction rate coefficients ($k_{i,x}$).

$$TCRE_x = \sum E_i\, k_{i,x} \qquad\qquad (3)$$

This determines approximately the relative role of the compounds or compound classes in local OH, and $O_3$ chemistry. The reaction rate coefficients are listed in Table 1. When available, temperature-dependent rate coefficients have been used. When experimental data was not available, the reaction coefficients have been estimated with the AopWin™ module of the EPI™ software suite (https://www.epa.gov/tsca-screening-tools/epi-suitetm-estimation-program-interface, EPA, U.S.A).

**3. Results and discussion**

**3.1 Weather patterns during the measurements**

According to the statistics of the Finnish Meteorological Institute, the weather conditions in Finland were close to normal during the growing season in the years the measurements were carried out. The main features of the weather patterns are characterised here briefly to provide an average estimate of the conditions in the measurement years compared with the long-term average (previous 30 year period) conditions in Finland.

In 2011, the spring was early and warm. Thermal spring (mean daily temperature above 0°C) started in the whole country during the first few days of April. The average temperatures in central Finland were 2-3 degrees higher than the normal long-term average temperatures. The precipitation in April was about 70 % of the long-term average, and even a little less in central Finland.
The same pattern continued in May, with slightly higher temperatures than the normal long-term average. Towards the end of the month the weather turned more unstable, with more rains and cooler night temperatures. The average temperature in June was a little over two degrees higher than the normal long-term average, and there were some intense thunderstorms.

In 2014, the weather conditions in May were quite typical, with the average temperatures close to the long-term average values in all parts of the country. The month started with temperatures cooler than the long-term average, and the cool period continued for about three weeks. After the cool period the weather became warmer with a south-eastern air flow, and hot (over 25°C) air temperatures were observed in southern and central parts of the country. Towards the end of May, cooler air spread over the country from the northeast, and the temperature drops could be high in eastern Finland. May was also characterised with precipitation, especially in eastern Finland. June started with a warm spell, but towards the end the weather was much cooler, with the average temperatures 1 to 2 degrees lower than the long-term average. The precipitation was regionally quite variable in June, the amount could be doubly the long-term

average in some areas, while the amounts were only half of it in many places in central Finland. July was much warmer
than the long-term average temperatures, especially in western Finland and in Lapland. July also had very little rain.
In 2015, the June average temperatures were 1 to 2 degrees below the long-term averages, especially in the western
parts of central Finland, and southern Lapland. There were also more rain showers than normally. In July the cold spell
and rainy days continued, with the average temperatures below the long-term averages, especially in the eastern parts
of the country. Highest precipitation rates were measured in the southern and western coastal regions, and in the eastern
parts of the country. In August the warmth returned after two cooler months, with average temperatures 1 to 2 degrees
above the long-term average values. August also had very little rain, except for some parts in eastern Finland and in
Lapland.
The observed mean temperature and precipitation amounts at the Juupajoki weather station in Hyytiälä during each
measurement month in 2011, 2014, and 2015 are shown in Table 2.

## 210 3.2 Variability of the VOC emissions

Seasonal mean emission rates of isoprene, 2-methyl-3-buten-2-ol (MBO), MTs and SQTs are presented in Table 3
and Fig 2. Typical diurnal variations of the most abundant compounds for each season are shown in Fig. 3. Since
most of the emission rates of the measured compounds were higher in late summer than in early summer, we
calculated the spring (April and May), early summer (June to mid-July) and late summer (late July and August) mean
emissions separately. This describes the emission rate changes better than monthly means.
Isoprene emission rates were low in spring and early summer, but increased in August. In spring emission rates were
below detection limit most of the time and early and late summer means were $1.3\pm3.7$ and $6.0\pm12$ ng g(dry weight)$^{-1}$
h$^{-1}$, respectively. The highest daily maxima isoprene emissions were about 70-80 ng g(dw)$^{-1}$ h$^{-1}$, but usually they
remained below 20 ng g(dw)$^{-1}$ h$^{-1}$. Our measured values (Table 3) match very well with the measurements by
Bourtsoukidis et al (2014b) who report season medians varying from 1.6 ng g(dry weight)$^{-1}$ h$^{-1}$ in autumn to 3.7 ng
g(dry weight)$^{-1}$ h$^{-1}$ in spring. However, while the highest emission rates were measured in late summer in the present
study, Bourtsoukidis et al. (2014b) found highest emission rates in spring.
MT emission rates were below 50 ng g(dw)$^{-1}$ h$^{-1}$ most of the time in April, May and still in the beginning of June for
every measurement year, below 50 ng g(dw)$^{-1}$ h$^{-1}$ most of the time. At the end of June the MT emission rates started to
increase (about 30 %) to the level where they remained until the end of August, the daily maxima or their sum
remaining below 300 ng g(dw)$^{-1}$ h$^{-1}$. In comparison with the study by Bourtsoukidis et al. (2014b), MT emission rates
in Finland are four to ten times lower than those measured in Germany and their seasonal cycles are different. As with
isoprene, they measured the highest MT emission rates during spring, whereas our highest emissions take place late
summer. Median seasonal values reported by them are 203.1, 136.5 and 80.8 ng g(dw)$^{-1}$ h$^{-1}$ for spring, summer and
autumn, respectively. Our averages are 8, 21 and 28 ng g(dw)$^{-1}$ h$^{-1}$ for spring, early summer and late summer,
respectively (Table 3).
A substantial change in the emission patterns took place at the end of July, when SQT emission rates increased up to
3-4 times higher than the MT emission rates at the same time (Table 3). Such a change in emissions was not observed
in the study by Bourtsoukidis et al. (2014b). ). Instead of late summer increase, they observed again highest emissions
already during the spring (118.6 and 64.9 ng g(dw)$^{-1}$ h$^{-1}$ in spring and summer, respectively) after which emissions
significantly declined. Moreover, they report that MTs dominated the Norway spruce emissions through the entire
measuring period (April-November), SQT emission rates being equal to MT emission rates during spring, but only
about half of MT emission rates during summer and about 20 % during autumn. One potential explanation for such a
different seasonality and emission strengths may lie in the differences between site specific factors such as soil moisture
conditions, local climate (winter in Germany is much milder and the trees do not face as dramatic change as in Finland
when winter turns to spring), stand age or stress factors. The tree measured in Germany was much older (about 80
years). In a boreal forest, late summer normally is the warmest and most humid season favouring high emissions, as
was also the case in our study periods. On the contrary, in central Germany July was relatively cold and wet, and
according to the authors, reduced emissions were therefore not surprising (Boutsourkidis et al 2014b).
Another interesting feature can be seen in the specified emission rates of different compounds. In the present study the
main SQT in spruce emissions was β-farnesene. About 50% of the SQT emission consisted of β-farnesene and its
maximum emission rate (155 ng g(dw)$^{-1}$ h$^{-1}$) was measured on the afternoon of 31 July 2015. Two other identified
SQTs were β-caryophyllene and α-humulene. There were two more SQTs, which also contributed significantly to the
total SQT emission rates, but since no calibration standards were available for these , their quantification is only
tentative. Linalool emissions increased simultaneously with SQT emissions (Fig. 2) reaching maximum concentrations
during late summer in August, in the same way as was previously observed in the measurements of Scots pine
emissions in the same forest in southern Finland (Hakola et al., 2006), where emissions were found to increase late
summer concomitant with the maximum concentration of the airborne pathogen spores, and Hakola et al. (2006)
suggested a potential defensive role of the conifer linalool and SQT emissions. Several other reports point to similar
correlations between SQT (in particular β-farnesene) and oxygenated MTs such as linalool emissions and biotic
stresses in controlled experiments. For example, increases in farnesene, methyl salicylate (MeSA) and linalool
emissions were reported to be an induced response by Norway spruce seedlings to feeding damage by mite species
(Kännaste et al. 2009), indicating that their biosynthesis might prevent the trees from being damaged. Interestingly,
the release of β-farnesene seemed to be mite specific and attractive to pine weevils, whereas linalool and MeSA were
deterrents. Blande et al. (2009) discovered pine weevil feeding to clearly induce the emission of MTs and SQTs,
particularly linalool and (E)-β-farnesene, from branch tips of Norway spruce seedlings, Also, in a licentiate thesis of
Petterson (2007) linalool and β-farnesene were shown to be emitted due to stress. The emissions from Norway spruce
increased significantly after trees were treated with methyljasmonate (MeJA). Martin et al (2003) discovered that MeJA
triggered increases in the rate of linalool emission more than 100-fold and that of SQTs more than 30-fold. Emissions
followed a pronounced diurnal rhythm with the maximum amount released during the light period, suggesting that they
are induced de novo after treatment. Our study shows that such major changes in emission patterns can also occur in
trees in field conditions, and without any clear visible infestations or feeding, indicating that they probably are systemic
defence mechanisms rather than direct ones (Eyles et al 2010).
In 2015 we measured also acetone/propanal and $C_4$-$C_{10}$ aldehyde emission rates. The total amount of these measured
carbonyl compounds was comparable to the amount of MTs (Table 3) although with our method it was not possible to
measure emissions of the most volatile aldehydes, formaldehyde and acetaldehyde, which are also emitted from trees
in significant quantities (Cojocariu et al., 2004, Koppmann and Wildt, 2007; Bourtsoukidis et al., 2014b). In summer
2015 the carbonyl compounds consisted mainly of acetone (30 %), and the shares for the other compounds were as
follows: nonanal (21%), decanal (17%), heptanal (14%), hexanal (10%) and pentanal (5%). The shares of butanal and
octanal were less than 2% each. The seasonal mean values are shown in Table 3. Aldehydes with shorter carbon
backbones (butanal, pentanal, hexanal) have higher emissions in early summer like most MTs, while aldehyges with
longer carbon backbones (heptanal, octanal, nonanal, decanal) have higher emissions in late summer similarly to SQTs.
Diurnal variability of the emission rates of MT and SQT, acetone/propanal and larger aldehydes are shown in Fig.2.
They all show similar temperature dependent variability with maxima during the afternoon and minima in the night.
The SQT daily peak emissions were measured two hours later than MT and aldehyde peaks.

**3.3 Tree to tree variability in emission pattern**

When following the emission seasonality, we discovered that the MT emission patterns were somewhat different
between the two trees measured. The tree measured in 2011 (tree 1) emitted mainly α-pinene in May, whereas the tree
measured in 2014 and 2015 (tree 2) emitted mainly limonene in May (Table 4). As summer proceeded the contribution
of limonene emission decreased in both trees and the share of α-pinene increased in tree 2. The species specific Norway
spruce emissions have been measured earlier at least by Hakola et al. (2003) and Bourtsoukidis et al. (2014a). The
measurements by Hakola et al. covered all seasons, but only a few daytime samples for each season, whereas the
measurements by Bourtsoukidis et al. covered three weeks in September-October in an Estonian forest. The main MTs
detected in the Estonian forest were α-pinene (59 %) and 3-carene (26 %), but also camphene, limonene, β-pinene and
β-phellandrene were detected. In the study by Hakola et al. (2003) the MT emission composed mainly of α-pinene, β-
pinene, camphene and limonene, but only very small amounts of 3-carene were observed, similarly to the present study.
This raises a question whether spruces would have different chemotypes in a similar way as Scots pine has (Bäck et
al., 2012).
In order to find out how much variability there was between the trees in monoterpene emission pattern, we conducted
a study in June in 2014, where we made qualitative analysis from six different spruces growing in a same area (labelled
as tree 3 - tree 8). The results for MT emissions are shown in Figure 4. SQT emissions were not significant at that time
(about 1 ng g(dw)$^{-1}$ h$^{-1}$).   As expected, the MT emission pattern of the trees was quite different; terpinolene was one
of the main MT in the emission of four trees whereas tree 3 emitted only 3% terpinolene. Also limonene and camphene
contributions were varying from few percent to about third of the total MT emission. All the measured trees emitted
rather similar proportions of α- and β-pinene. The shares of myrcene, β-pinene and 3-carene were low in every tree.
Since different MTs react at different rates in the atmosphere (Table 1), the species specific measurements are
necessary when evaluating MTs influence on atmospheric chemistry. Currently, air chemistry models very often use
only single branch measurements and this can lead to biased results when predicting product and new particle
formation. This study and the study of Scots pine emissions by Bäck et al. (2012) show that species specific
measurements are necessary, but also that flux measurements are more representative than branch scale emission
measurements and averaging over larger spatial scale may be better suited for air chemistry models.
**3.4 Standard emission potentials**
The standard emission potentials were obtained by fitting the measured emission rates to the temperature dependent
pool emission algorithm (equation 2) and the light and temperature dependent algorithm (equations 3-5) described in
section 2.2). For the temperature dependent algorithm, the nonlinear regression was carried out with two fitted
parameters, yielding both the emission potentials and individual β coefficients for each compound group. With the
light and temperature dependent algorithm, only emission potentials were obtained. The compounds' emissions fitted
using the temperature dependent pool emission algorithm were the ones of the most abundant MT, SQT and the sum
of carbonyls for each season, while the analysis with the light and temperature dependent emission algorithm was
carried out for isoprene emissions. In the analysis, obvious outliers and other suspicious data were not included. The
excluded values typically were the first values obtained right after starting a measurement period, which might still
show the effects of handling the sample branch. The isoprene emissions obtained in 2011 were not taken into account
in the analysis as they were not properly collected on the cold trap. This was fixed in 2014 and 2015 by changing the
adsorbent material. An approach with a hybrid algorithm, where the emission rate is described as a function of two
source terms, de novo synthesis emissions and pool emissions, was also tested. However, the results were not
conclusive.
The standard emission potentials of isoprene, the selected MT and SQT, acetone and $C_4$-$C_6$ aldehyde sums are presented
in Table 5. Emission potentials are given as spring, early summer, and late summer values. The coefficient of
determination ($R^2$) is also given, even though it is an inadequate measure for the goodness of fit in nonlinear models
(e.g. Spiess and Neumeyer, 2010). A more reliable parameter for estimating the goodness of fit is the standard error of
the estimate, which is also given.
The summertime emission potentials of MT and SQT reflect the typical behaviour of the temperature variability in
summer, with low emissions in spring and high emissions in the higher temperatures of late summer. The variability
of the emission potential during the growing season and between the individual compounds is large. In late summer
limonene and α-pinene had the highest MT emission potentials. SQT exhibit a similar behavior as monoterpene
emission potentials with very low springtime and early summer emission potentials while the late summer emission
potential is high. In a review by Kesselmeier and Staudt (1999) the reported standard emission potentials (30°C, 1000
μmol m$^{-2}$ s$^{-1}$) of Norway spruces for monoterpenes vary from 0.2 to 7.8 μg g(dry weight)$^{-1}$ h$^{-1}$ and in a study by
Bourtsoukidis et al. (2014b) mean emission potential of Norway spruce was 0.89 μg g(dry weight)$^{-1}$ h$^{-1}$ for all data
(spring, summer, fall). Our standardized MT emission potentials are lower than earlier reported values being 0.1 μg
g(dry weight)$^{-1}$ h$^{-1}$ during late summer, when they were at their highest.
This is the first time we have applied fitting the traditional temperature-based emission potential algorithms to
measured carbonyl emissions, and based on the spruce emission results, the approach appears to be applicable also on
these compounds. The best fit was obtained with the temperature dependent algorithm. The temporal variability of the
emission potential was similar to MT- and SQTs. Unfortunately, acetone/propanal and C4-C10 aldehyde measurements
were only carried out during the last measurement campaign, but the emission pattern possibly indicates a midsummer
maximum, because emissions were clearly identified in June, and already decreasing in late July-August. The isoprene
emissions, fitted with the light and temperature emission algorithm, also reflect the light/temperature pattern of
summer, with low emissions in spring and high emissions in late summer.
In late summer when isoprene emissions were a bit higher the emission model fits the data better and the emission
potential for isoprene was 56.5 ng g(dry weight)$^{-1}$h$^{-1}$. In a review by Kesselmeier and Staudt (1999) the reported
standard emission potentials (30°C, 1000 μmol m$^{-2}$ s$^{-1}$) of isoprene vary from 0.34 to 1.8 μg g(dry weight)$^{-1}$ h$^{-1}$. Our
standardized late summer mean (56.5 ng g(dry weight)$^{-1}$ h$^{-1}$) is much lower than these earlier reported values.
**3.5 Relative reactivity of emissions**
In summer in ambient air at this site most of the known OH reactivity (which is ~50 % of the total measured OH
reactivity) is coming from the VOCs (Sinha et al. 2010; Nölcher et al. 2012). Other trace gases (NO$_x$, CO, O$_3$, CH$_4$)
have a lower contribution. Of these VOCs, aromatic hydrocarbons have only minor contribution compared to the
terpenoids (Hakola et al. 2012). In these ambient air studies contribution of SQTs has been much lower than MTs, but
those results are misleading, since lifetimes of most SQTs are so short that they can not be detected in ambient air and
estimation of their contribution to the local reactivity is possible only directly from the emissions. Here we studied the
relative role of different BVOCs to the reactivity of Norway spruce emissions.
The relative contribution from each class of compounds to the total calculated OH and O$_3$ reactivity of the emissions
$TCRE_{OH}$ and $TCRE_{O3}$, respectively, is depicted in Fig. 5. Nitrate radicals are likely to contribute also significantly to
the reactivity, but since the reaction rate coefficients were not available for the essential compounds like β-farnesene,
the nitrate radical reactivity is not shown. SQT are very reactive towards ozone and they clearly dominate the ozone
reactivity. Isoprene contribution is insignificant all the time towards ozone reactivity, but it contributes 20-30 % of OH
reactivity, although the emission rates are quite low. SQT dominate also OH reactivity during late summer due to their
high emission rates, but early summer MT contribution is equally important.  Contribution of acetone to the $TCRE_{OH}$
was very small (~0.05% of total reactivity), but reactivity of C$_4$-C$_{10}$ aldehydes was significant, averagely 15% and
sometimes over 50% of the $TCRE_{OH}$. Of the aldehydes decanal, nonanal and heptanal had the highest contributions. It
is also possible to measure total OH reactivity directly and experimental total OH reactivity measurements by Nölscher
et al. (2013) showed that the contribution of SQTs in Norway spruce emissions in Hyytiälä was very small (~1%). This
is in contradiction to our measurements, where we found very high share of SQTs (75% in late summer). Nölscher et
al. (2013) found also very high fraction of missing reactivity (>80%) especially in late summer. Their measurements
covered spring, summer and autumn. Emissions of C$_4$-C$_{10}$ aldehydes, which were not studied by Nölscher et al. (2013)
could explain part of the missing reactivity.

**4 Conclusions**

Norway spruce VOC emissions were measured in campaigns in 2011, 2014 and 2015. Measurements covered altogether 14 spring and summer weeks. The measured compounds included isoprene, MT and SQT and in 2015 also acetone and $C_4$-$C_{10}$ aldehydes. MT and SQT emission rates were low during spring and early summer. MT emission rates increased to their maximum at the end of June and declined a little in August. A significant change in SQT emissions took place at the end of July, when SQT emissions increased substantially. The seasonality is different from that observed earlier in Germany (Bourtsoukidis et al. 2014b). There Norway spruce emissions (isoprene, MT, SQT) were highest in spring and declined thereafter. The difference in seasonality can be due to different ages of the measured trees (10-15 years in the current study, 80 years in Bourtsoukidis et al. 2014b), different climate or different stress factors. These same factors can also cause lower emission rates measured now in comparison with other studies. The effect of age to the emission potentials should be studied.

In August SQT were the most abundant group in the emission, β-farnesene being the most dominant compound. SQT emissions increased simultaneously with linalool emissions and these emissions were suggested to be initiated due to stress effects. To our knowledge this is the first time when β-farnesene and linalool emissions have been shown to increase simultaneously in natural conditions, although they have been shown to increase in the emissions together due to stress effects. Of the measured compounds, SQTs had highest impact on local $O_3$ and OH chemistry. This clearly shows the importance of considering also SQTs in atmospheric studies in boreal environment.

Acetone and $C_4$-$C_{10}$ aldehyde emissions were highest in July, when they were approximately at the same level as MT emissions. $C_4$-$C_{10}$ aldehydes contributed as much as MT to the OH reactivity during late summer, but early summer only about half of the MT share in early summer. This demonstrates that also emissions of other BVOCs than the traditionally measured terpenoids are important and should be included in atmospheric studies.

The MT emission pattern varies a lot from tree to tree. During one afternoon in June we measured emission pattern of six different trees growing near each other and especially the amounts of terpinolene, camphene and limonene were varying. Due to inconsistent emission pattern the species specific emission fluxes on canopy level should be conducted in addition to the leaf level measurements for more representative measurements. However, only leaf level measurements produce reliable SQT data.

**Acknowledgements**

The financial support by the Academy of Finland Centre of Excellence program (project no 272041) and Academy Research Fellow program (project no 275608) are gratefully acknowledged. The authors thank Dr. Juho Aalto for determining the specific leaf area of the needles.

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

 **Table 1: OH and O3 reaction rate coefficients used in reactivity calculations.**

| Species | $k_{OH}$ (cm$^3$ s$^{-1}$) | Reference | $k_{O3}$ (cm$^3$ s$^{-1}$) | Reference |
|---|---|---|---|---|
| Isoprene | $2.7 \cdot 10^{-11} \cdot e^{390/T}$ | Atkinson et al. (2006)[a] | $1.03 \cdot 10^{-14} e^{-1995/T}$ | Atkinson et al. (2006)[a] |
| 2-Methyl-3-buten-2-ol | $6.3 \cdot 10^{-11}$ | Atkinson et al. (2006)[a] | $1.0 \cdot 10^{-17}$ | Atkinson et al. (2006)[a] |
| α-Pinene | $1.2 \cdot 10^{-11} \cdot e^{440/T}$ | Atkinson et al. (2006)[a] | $8.05 \cdot 10^{-16} \cdot e^{-640/T}$ | IUPAC[b] |
| Camphene | $5.33 \cdot 10^{-11}$ | Atkinson et al. (1990a) | $6.8 \cdot 10^{-19}$ | IUPAC[b] |
| Sabinene | $1.17 \cdot 10^{-10}$ | Atkinson et al. (1990a) | $8.2 \cdot 10^{-17}$ | IUPAC[b] |
| β-Pinene | $1.55 \cdot 10^{-11} \cdot e^{467/T}$ | Atkinson and Arey (2003) | $1.35 \cdot 10^{-15} \cdot e^{-1270/T}$ | IUPAC[b] |
| Myrcene | $9.19 \cdot 10^{-12} \cdot e^{1071/T}$ | Hites and Turner (2009) | $2.65 \cdot 10^{-15} \cdot e^{-520/T}$ | IUPAC[b] |
| 3-Carene | $8.8 \cdot 10^{-11}$ | Atkinson and Arey (2003) | $4.8 \cdot 10^{-17}$ | IUPAC[b] |
| p-Cymene | $1.51 \cdot 10^{-11}$ | Corchnoy and Atkinson (1990) | $< 5.0 \cdot 10^{-20}$ | Atkinson et al. (1990b) |
| Limonene | $4.2 \cdot 10^{-11} \cdot e^{401/T}$ | Gill and Hites (2002) | $2.8 \cdot 10^{-15} \cdot e^{-770/T}$ | IUPAC[b] |
| 1,8-Cineol | $1.11 \cdot 10^{-11}$ | Corchnoy and Atkinson (1990) | $< 1.5 \cdot 10^{-19}$ | Atkinson et al. (1990) |
| Linalool | $1.59 \cdot 10^{-10}$ | Atkinson et al. (1995) | $\geq 3.15 \cdot 10^{-16}$ | Grosjean and Grosjean (1998) |
| Terpinolene | $2.25 \cdot 10^{-10}$ | Corchnoy and Atkinson (1990)[a] | $1.6 \cdot 10^{-15}$ | IUPAC[b] |
| Bornylacetate | $1.39 \cdot 10^{-11}$ | Coeur et al. (1999) | - | |
| Longicyclene | $9.35 \cdot 10^{-12}$ | AopWin™ v1.92 | - | |
| Isolongifolene | $9.62 \cdot 10^{-11}$ | AopWin™ v1.92 | $1.0 \cdot 10^{-17}$ | IUPAC[b] |
| β-Caryophyllene | $2.0 \cdot 10^{-10}$ | Shu and Atkinson (1995) [a] | $1.2 \cdot 10^{-14}$ | IUPAC[b] |
| β-Farnesene | $1.71 \cdot 10^{-10}$ | Kourtchev et al. (2012) | $1.5 \cdot 10^{-12} \cdot e^{-2350/T}$ | IUPAC[b] |
| α-Humulene | $2.9 \cdot 10^{-10}$ | Shu and Atkinson (1995)[a] | $1.2 \cdot 10^{-14}$ | IUPAC[b] |
| Alloaromadendrene | $6.25 \cdot 10^{-11}$ | AopWin™ v1.92 | $1.20 \cdot 10^{-17}$ | AopWin™ v1.91 |
| Zingiberene | $2.87 \cdot 10^{-10}$ | AopWin™ v1.92 | $1.43 \cdot 10^{-15}$ | AopWin™ v1.91 |
| Acetone | $8.8 \cdot 10^{-12} \cdot e^{-1320/T} +$ $1.7 \cdot 10^{-14} \cdot e^{423/T}$ | Atkinson et al. (2006)[a] | - | |
| Butanal | $6.0 \cdot 10^{-12} \cdot e^{410/T}$ | Atkinson et al. (2006)[a] | - | |
| Pentanal | $9.9 \cdot 10^{-12} \cdot e^{306/T}$ | Thévenet et al. (2000) | - | |
| Hexanal | $4.2 \cdot 10^{-12} \cdot e^{565/T}$ | Jiménez et al. (2007) | - | |

| | | | | |
|---|---|---|---|---|
| Heptanal | $2.96 \cdot 10^{-11}$ | Albaladejo et al. (2002) | - | |
| Octanal | $3.2 \cdot 10^{-11}$ | AopWin™ v1.92 | - | |
| Nonanal | $3.6 \cdot 10^{-11}$ | Bowman et al. (2003) | - | |
| Decanal | $3.5 \cdot 10^{-11}$ | AopWin™ v1.92 | - | |

[a]IUPAC recommendation
[b]IUPAC Task Group on Atmospheric Chemical Kinetic Data Evaluation (http://iupac.pole-ether.fr).
**Table 2: Mean temperatures (°C) and rain amounts (mm) during each measurement month in Hyytiälä**.

| | 2011 | | 2014 | | 2015 | |
|---|---|---|---|---|---|---|
| | temp | rain | temp | rain | temp | rain |
| April | 4.5 | 17.4 | | | | |
| May | 9.3 | 44.3 | 9.4 | 57.4 | | |
| June | 15.8 | 65.3 | 11.8 | 94.8 | 11.9 | 81.5 |
| July | | | 18.6 | 44.1 | 14.6 | 86.7 |
| August | | | | | 15.2 | 12.6 |


**Table 3: Seasonal mean emission rates of isoprene, 2-methylbutenol (MBO), MT, SQT, acetone and C4-C10 carbonyls in ng g(dw)$^{-1}$ h$^{-1}$. "na" means that the compounds were not included in the analysis. Spring is April-May, early summer 1.6-15.7 and late summer 16.7-31.8. bdl = below detection limit. Values are averages and standard deviations for the three measurement years (2011, 2014, 2015). Other SQT = sum of all other SQTs in emissions. The number of the measurements each season is in parentheses.**

| | average spring | stdev spring | average early summer | stdev early summer | average late summer | stdev late summer |
|---|---|---|---|---|---|---|
| isoprene | | | 1.3 | 3.7 | 6.0 | 12 |
| MBO | | | 2.1 | 4.2 | 2.4 | 3.8 |
| | | | | | | |
| Camphene | 1.1 | 1.8 | 2.9 | 4.4 | 3.8 | 4.1 |
| 3-Carene | 0.3 | 0.7 | 1.1 | 1.7 | 0.9 | 0.6 |
| p-cymene | 0.3 | 0.6 | 0.9 | 1.8 | 0.5 | 0.5 |
| Limonene | 2.7 | 3.4 | 6.1 | 12.2 | 7.7 | 9.5 |
| Myrcene | 0.2 | 0.4 | 1.7 | 3.7 | 3.9 | 5.1 |
| α-Pinene | 2.1 | 3.4 | 5.8 | 11.1 | 9.6 | 11 |
| β-Pinene | 1.0 | 2.2 | 1.8 | 6.2 | 0.9 | 1.1 |
| Sabinene | 0 | 0.1 | 0.5 | 1.5 | 0.9 | 1.6 |
| terpinolene | 0 | 0.2 | 0.1 | 0.4 | 0.3 | 0.9 |
| | | | | | | |
| bornylacetate | 0 | 0.2 | 0.5 | 2.0 | 1.1 | 2.1 |
| 1,8-Cineol | 0.7 | 0.7 | 2.1 | 3.9 | 1.8 | 2.2 |
| linalool | na | | 1.4 | 2.2 | 7.9 | 12 |
| | | | | | | |
| β-caryophyllene | 0 | 0 | 0.4 | 2.1 | 7.2 | 5.9 |
| β-farnesene | 0 | 0 | 1.1 | 4.3 | 42 | 29 |
| other SQT | 0.1 | 0.4 | 1.4 | 4.7 | 35 | 30 |
| | | | | | | |
| Acetone | na | | 17 | 11 | 17 | 9.0 |
| Butanal | na | | 2.0 | 0.7 | 0.3 | 0.3 |
| Pentanal | na | | 4.1 | 1.1 | 2.4 | 0.9 |
| Hexanal | na | | 5.0 | 3.0 | 4.9 | 2.1 |
| Heptanal | na | | 5.2 | 1.2 | 7.5 | 2.4 |
| Octanal | na | | 0.3 | 0.1 | 0.4 | 1.1 |
| Nonanal | na | | 6.3 | 2.3 | 9.9 | 4.5 |
| Decanal | na | | 5.6 | 2.3 | 7.4 | 3.8 |







**Table 4: Average monthly abundances (%) of emitted MTs. T1 (tree1) includes 2011 and T2 2014 and 2015 measurements. The number of the measurements each month is in parentheses**.

|  | α-Pinene | Camphene | Sabinene | β-Pinene | Myrcene | Δ³-Carene | p-Cymene | Limonene | Terpinolene |
|---|---|---|---|---|---|---|---|---|---|
| April, T1 (160) | 34 | 19 | 0 | 18 | 1 | 5 | 6 | 18 | 0 |
| May, T1 (48) | 59 | 9 | 1 | 7 | 1 | 1 | 9 | 10 | 3 |
| June, T1 (34) | 7 | 25 | 16 | 0 | 34 | 3 | 9 | 4 | 0 |
| | | | | | | | | | |
| May, T2 (129) | 16 | 11 | 0 | 10 | 5 | 5 | 2 | 51 | 0 |
| June, T2 (396) | 27 | 15 | 0 | 15 | 5 | 5 | 4 | 29 | 0 |
| July, T2 (128) | 32 | 15 | 2 | 5 | 7 | 5 | 2 | 27 | 1 |
| Aug T2 (134) | 34 | 11 | 3 | 3 | 15 | 3 | 1 | 29 | 1 |





Table 5: Standard (30 °C) MT, SQT, acetone and $C_4-C_{10}$ aldehyde emission potentials obtained in 2011, 2014 and
2015. For isoprene the standard (1000 µmol photons $m^{-2}$ $s^{-1}$, 30 °C) emission potentials are from the 2015 campaign.
The standard emission potential $E_S$ and the $\beta$ coefficient are given with the standard error of the estimate (StdErr, in
parenthesis). R squared and the number of measurements (N, in parenthesis). The fits were made for the spring
(April - May), early summer (June – mid July) and late summer (late July – August) periods.

| | Es (StdErr) ng/g(dw)*h | β K-1 (StdErr) | R2 (N) |
|---|---|---|---|
| **Spring** | | | |
| α-pinene | 11.6 (0.7) | 0.097 (0.006) | 0.423 (331) |
| camphene | 2.5 (0.4) | 0.045 (0.009) | 0.071 (323) |
| β-pinene | 1.9 (0.2) | 0.044 (0.007) | 0.119 (324) |
| myrcene | 0.6 (0.1) | 0.010 (0.011) | 0.007 (157) |
| limonene | 5.0 (0.8) | 0.032 (0.008) | 0.049 (321) |
| other MT | 2.9 (0.2) | 0.085 (0.005) | 0.433 (329) |
| β-caryophyllene | 0.2 (0.1) | 0.018 (0.059) | 0.026 (6) |
| β-farnesene | - | - | - (0) |
| other SQT | 0.7 (0.3) | 0.046 (0.029) | 0.029 (72) |
| | | | |
| **Early summer** | | | |
| α-pinene | 14.1 (1.0) | 0.058 (0.006) | 0.145 (489) |
| camphene | 7.0 (0.3) | 0.060 (0.004) | 0.230 (492) |
| β-pinene | 5.2 (0.6) | 0.062 (0.010) | 0.076 (426) |
| myrcene | 5.8 (0.3) | 0.078 (0.005) | 0.326 (356) |
| limonene | 16.7 (0.9) | 0.069 (0.005) | 0.239 (497) |
| other MT | 7.0 (0.3) | 0.074 (0.004) | 0.385 (499) |
| β-caryophyllene | 4.8 (1.3) | 0.018 (0.019) | 0.023 (54) |
| β-farnesene | 6.9 (1.8) | 0.012 (0.018) | 0.007 (90) |
| other SQT | 6.2 (0.7) | 0.055 (0.010) | 0.087 (238) |
| acetone | 50.8 (7.2) | 0.066 (0.010) | 0.362 (71) |
| aldehydes | 59.1 (4.4) | 0.043 (0.005) | 0.503 (71) |
| | | | |
| **Late summer** | | | |
| isoprene | 56.5 (4.2) | | 0.473 (70) |
| α-pinene | 39.3 (4.1) | 0.153 (0.017) | 0.359 (163) |
| camphene | 7.7 (1.2) | 0.064 (0.016) | 0.094 (161) |
| β-pinene | 2.5 (0.3) | 0.075 (0.015) | 0.160 (120) |
| myrcene | 21.1 (2.0) | 0.191 (0.019) | 0.476 (154) |
| limonene | 32.3 (3.6) | 0.155 (0.018) | 0.336 (163) |
| other MT | 9.9 (1.1) | 0.133 (0.016) | 0.298 (153) |
| β-caryophyllene | 11.0 (1.2) | 0.020 (0.010) | 0.032 (129) |
| β-farnesene | 76.9 (7.5) | 0.060 (0.010) | 0.183 (162) |
| other SQT | 67.3 (8.2) | 0.059 (0.013) | 0.132 (157) |
| acetone | 31.8 (2.2) | 0.061 (0.007) | 0.313 (163) |
| aldehydes | 36.8 (3.0) | 0.008 (0.007) | 0.009 (163) |


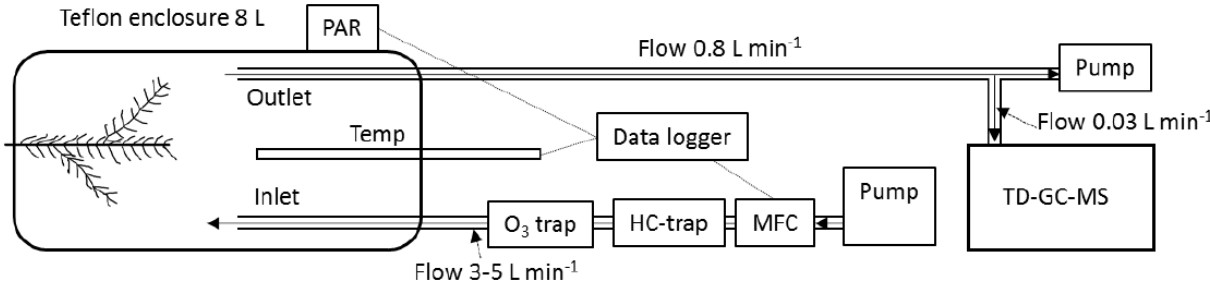

**Figure 1: The emission measurement set-up**

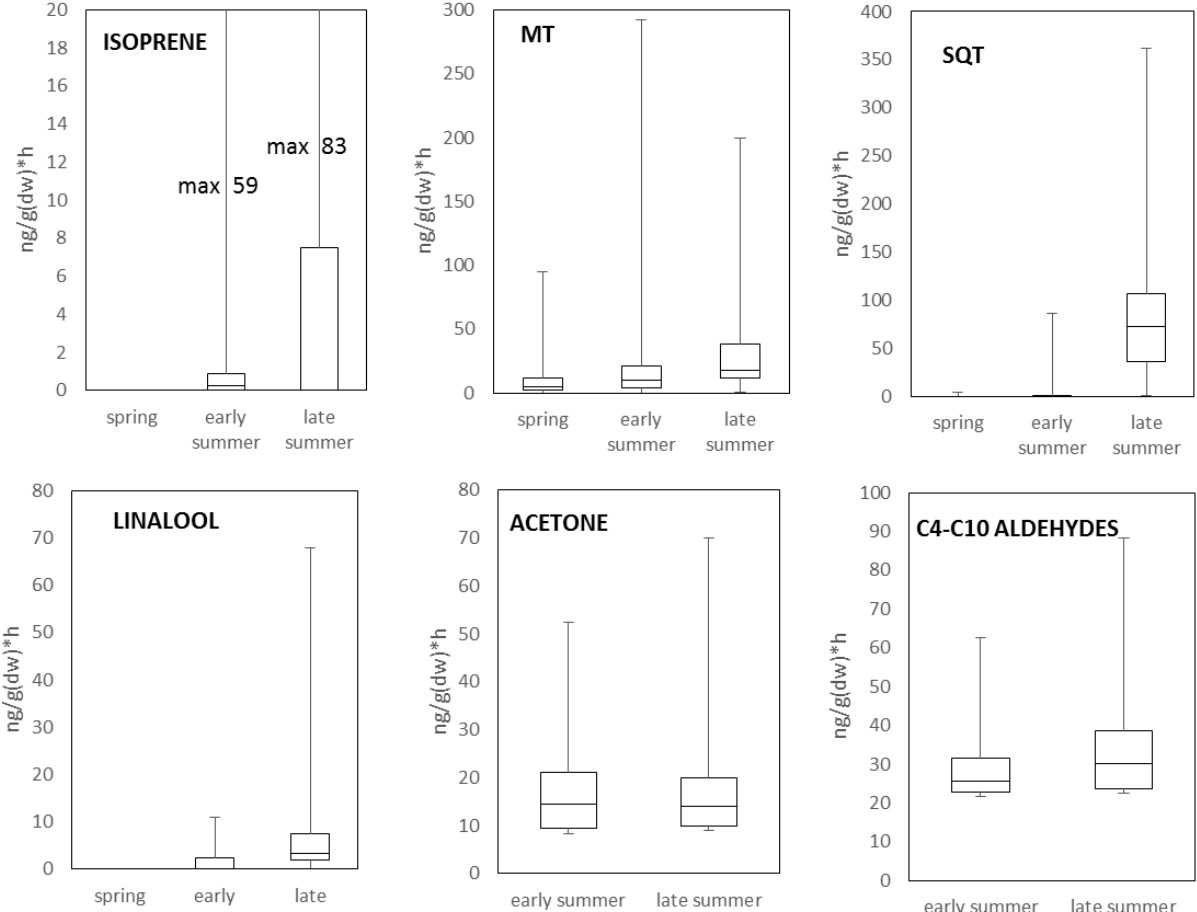

**Figure 2: Season mean box and whisker plots of isoprene, MT, SQT, acetone, C4-C10 aldehydes (butanal,**
**pentanal, hexanal, heptanal, octanal, nonanal and decanal) and linalool. Boxes represent second and third**
**quartiles and vertical lines in the boxes median values. Whiskers show the highest and the lowest observations.**

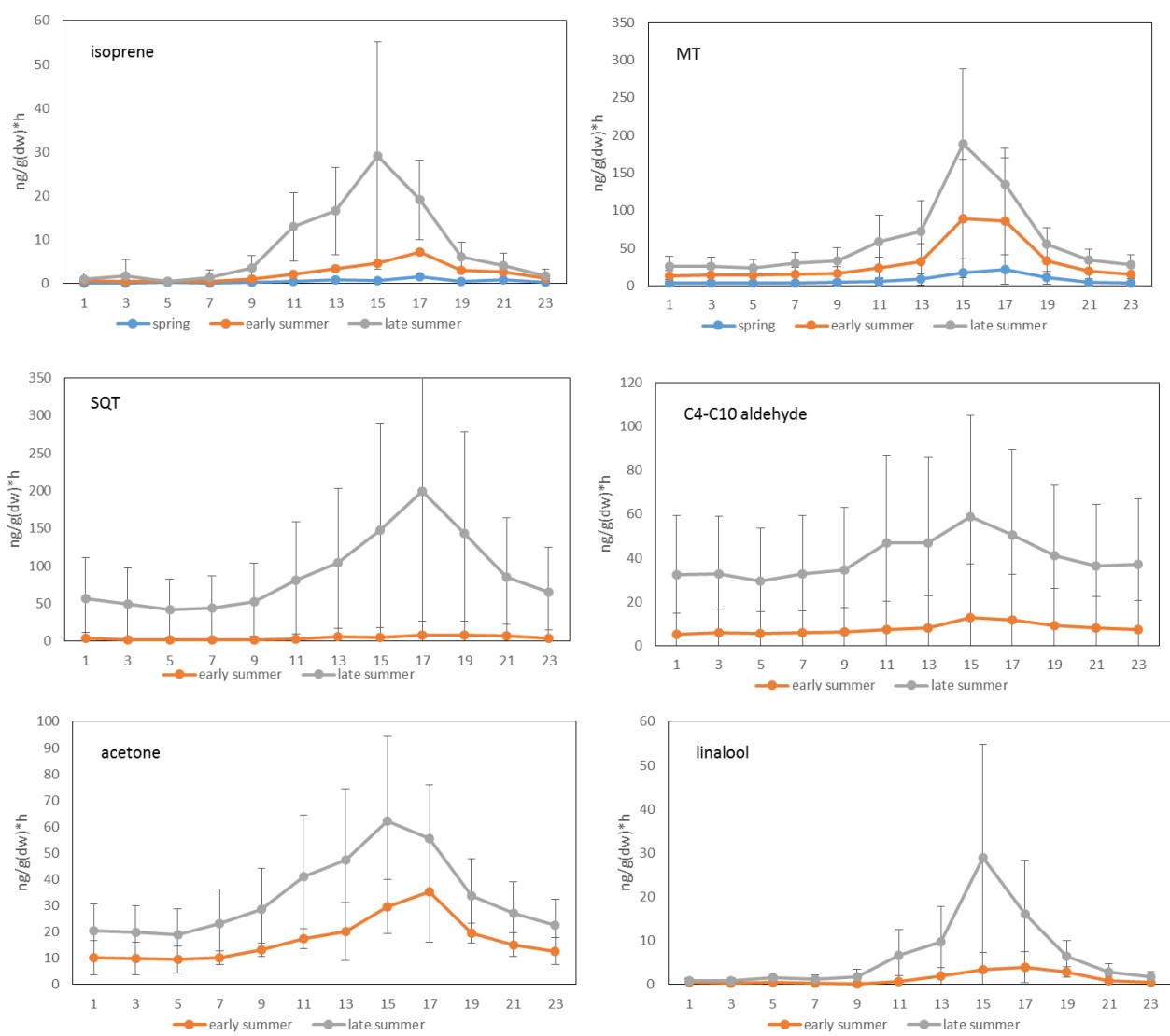


**Figure 3: Mean diurnal variations of different compound groups in each season. Spring refers to April and May, early summer June-mid July, late summer mid July-August. Aldehydes are sum of all C$_4$-C$_{10}$ aldehydes (butanal, pentanal, hexanal, heptanal, octanal, nonanal and decanal).**







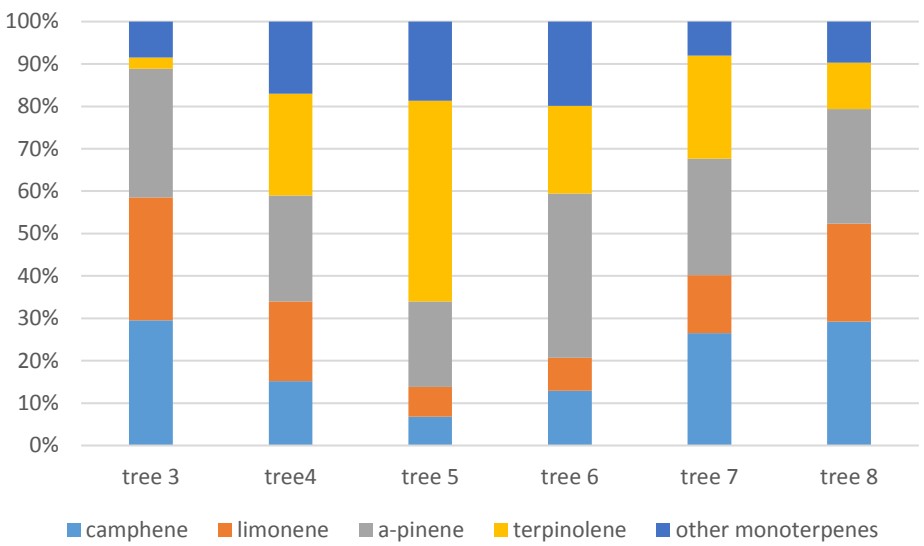


**Figure 4: Relative abundances of emitted monoterpenes in six different spruce individuals on 24 June 2014.**

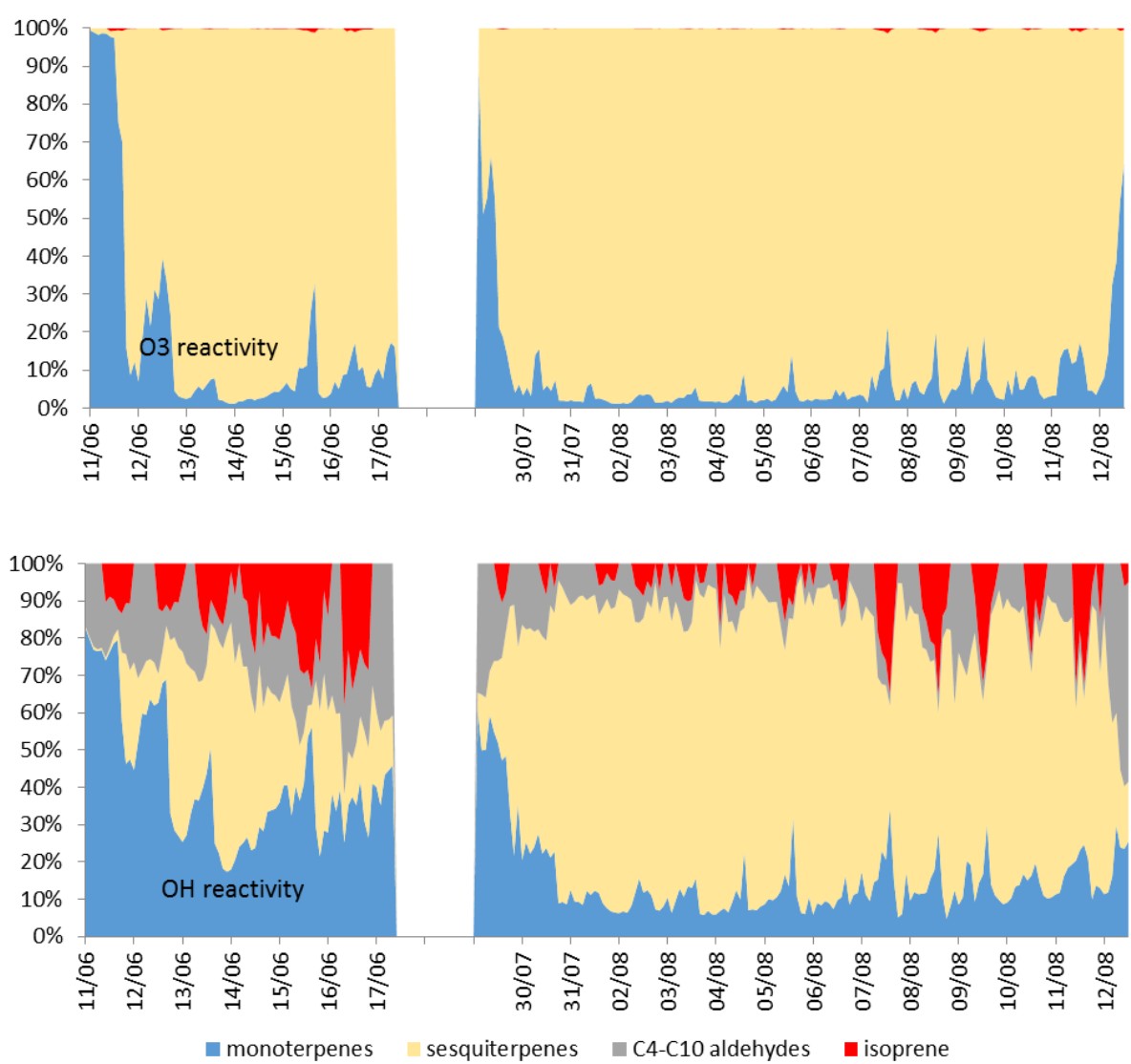

**Figure 5. Relative O₃ and OH reactivity of emissions for two periods in early and late summer 2015. The**
**compounds and reaction coefficients used for reactivity calculations are presented in Table 1.**
