# Peer review of "Terpenoid and carbonyl emissions from Norway spruce in Finland during the growing season"

_Atmospheric Chemistry and Physics, 2016_

## Referee Comment (RC1) · Anonymous Referee #2 · 31 Oct 2016

Review of Terpenoid, acetone, and aldehyde emissions from Norway spruce.

This study provides new information about the emission patterns of isoprene, monoterpenes, sesquiterpenes, acetone and c4-c10 carbonyl compounds in Norway spruce, one of the dominant species from boreal ecosystems. The manuscript reads well and it is providing new information that it is interesting for the scientific community and under the scope of ACP. Therefore, I accept it for publication; however, some requisites must be fulfilled before.

Abstract

Please state objectives and conclusions

Introduction

[Figure]

It is missing an overview on the boreal forest. Norway spruce has only been mentioned once. Please rewrite the introduction taking into account a better explanation of the boreal ecosystem and the role that VOC emissions have in such ecosystems

What is the objective? Very few data on emissions? I suppose there is another rationale, please state.

Page 1, Line 14: please insert references that show forested boreal emissions of mt, sqt and OVOCs. Page 1, Line 16: contribute to the increase of methane lifetime? How? Please explain. Page1, Line 26: You say in addition to isoprene, but is the first time you mention isoprene. Please expand. Page 1, Line 28: please give an example of saturated aldehydes.

Methods

The methods sections needs considerable attention. The measurement times and tress are expressed in a confusing manner, and better explanation of sampling must be given. Another important issue is the comparison among trees.

The tree measured in 2011 was different from the tree measured in 2014 and 2015, therefore I think they are not comparable as different processes such as age or different climatic conditions may come into play. This different tree should be removed from the comparison. Not necessarily from the study.

To begin with, a table with the different measured trees, years and techniques shall be stated. Furthermore, a better explanation of the cuvette used is needed, a picture of the setting will help the reader considerably. Is there a blank cuvette? How do you take into the possible effects of the cuvette on the branch used?

Page 2, line 50: You say here the samples were collected. What do you mean by that? Which samples? GC cartridges? You take samples from the outlet of the cuvette? Please specify. Page 2 line 55: you mention you have a thermometer inside the enclosure. What brand? Is this thermometer having a possible artefact effect? Page 2 line

56: how did you measure PPFD? Please include brand. Page 2 line 66: Please explain how the quantification of sabinene can be done suing the calibration curve of b-pinene. Page 2 line 70: please include the manufacturer of the calibration solutions. Page 3 line 72-75: here you say that in 2015 you were able to measure acetone and C4-C10 aldehydes. Then you say acetone was coeluted with propanal. Either you give a proof that you were able to properly calibrate acetone or you do not report acetone. In addition, please specify which C4-C10 aldehydes were you analysing. Page 3 line 93: there are more recent studies (Guenther et al., 2012) that suggest the slope value should be 0.1. Page 4 line 102: you say that you have used a temperature dependence for monoterpenes and a light and temperature dependence for isoprene. Please calculate also the temperature only dependence for isoprene and the light and temperature dependence for monoterpenes to conclude which is the best choice. Page 5 line 128-133: This part is confusing. You need to properly explain how the sampling was performed in the different years. So for this day on the 24th of June of 2014, you analysed 6 different spruces which then you compare to the 7th tree which is the one continuously measured in 2014. I don't understand how can they be comparable if the sampling is different (tree number 7 uses the Teflon cuvette via the dynamic flow through, whereas the other 6 tress were sampled with a Teflon bag. Did you have a blank? For how long were you sampling? I also noticed that for the cuvette tree the adsorbents are different than for the Teflon bag, and disturbances can be different, therefore I would not compare them together. You need to give tree numbers from the beginning of the methodology, so it is clearer to the reader. Furthermore, a more detailed information about sampling and how this is different to the main sampling is provided. I supposed these samples are analysed with the same instrument that is measuring cuvette air. Please state. Page 5 line 139: you say that when experimental data was not available you use this software. Then use it to estimate the reaction coefficient for b-farnense and nitrate, as you mention its importance (linked to page 9 line 288).

Results and discussion

3.1. Weather patterns during the measurements.

Here you need a graph showing the year to year variability. In the table you cant really see what are the changes. Furthermore, in table 2 you say that those are mean values, therefore is needed to use standard deviations. For the rain you must state the mm, is it mm per month?.

A better explanation of how do you consider the seasons is needed, therefore perhaps pointing in the new meteorological figure when it is spring, early summer, late summer, etc, can help and then you express in the text why.

Page 5 line 145: you say temperatures are exceptionally high and precipitation is extremely low. what is high, what is low. . .. This must be stated! Page 5 line 151: you mention a warm spell in June and a cold spell in July. Please show on new meteorological graph and explain what is a warm/cold spell. A clear and "based on meteorological data" election of the seasons must be stated. This can be added into the measurements table (i.e. year, tree, measurement technique (Cuvette, Teflon bag). . .)

3.2. Variability of VOC emissions

Page 6 line 157: what do you mean by early season? Page 6 line: 159-161: you cant compare the measurements of year 2011 to the measurements on 2014 and 2015 as they are different tree, so please only use 2014 and 2015 data for comparison. Furthermore, you present seasonal means and do no report of standard deviation. Please report standard deviations. Page 6 line 162: what is a low and a moderate emitter? Please report about values. Furthermore, this kind of information suits much better in the introduction. It would also be good to have a comparison among other high emitting species from the boreal region. Page 6 line 164: you say that you studies confirmed the low isoprene and moderate monoterpene emitters but the seasonal patterns were clearly different. Different to what? I have no indication of the seasonality of low isoprene and moderate monoterpene emitters, please mention and discuss. Page 6 line166: this is the only mention of MBO (apart from the methodology). I would skip

it or expand the explanation on MBO. Page 6 line 162-170: you report several emission rates but no uncertainty, please report. Page 6 line 171-172: you are still talking about monoterpenes and sesquiterpenes from your data, so this should go in the above paragraph. Table 3: In order to have an easier visibility of the data I prefer to see a bar graph of table 3, with uncertainties! Page 6 line 171-184: here you do a comparison with Bourtsoukidis et al., 2014b. This is a nice comparison. But I prefer that first you mention the why of your found seasonality in the boreal forest to then start stating the difference to the German forests and thus differences in emissions. Furthermore, you mention only a difference between 0-84 ng g(dw)-1h-1 for SQT in your study, and this is a big part of your results. Please expand your SQT results and then compare to other studies. Page 6 line 186: you say that the main sqt is b-farnense, can you comment about the other SQT measured? Page 6 line 189: this is an important result and statement, therefore please show a graph showing the Linalool and sqt increase together. Furthermore this can be another conclusion from you study. Page 7 line 193-207: This role should also be mentioned in the introduction. Furthermore, there has to be a better integration between the results from this study and the literature research. Page 7 line 209: If you cant measure most volatile aldehydes then it does not make sense to say that the amount of measured carbonyl compounds was comparable to the monoterpenes, as it is misleading. Page 7 line 212: Could you provide with mean values for the percentages? Was this percentage calculated from both early and late summer, or they were calculated separately? Page 7 line 213: you mention the possibility of bidirectional exchange when moist vegetation. Why? What is the link to your study? Please state. Figure 1: please include light as well to see the effect that light can have. Please remove/separate the graph from 2011 as it is not comparable to the other years as you were measuring a different tree. Please report as well standard deviations, name the compounds in the sum of C4-C10 aldehydes. If you were not able to give a proper explanation of the calibration for acetone, please remove from graph. In addition just a as help for the reader indicate which months comprehend the different selected seasons.

**3.3. Tree to tree variability in emission pattern**

It is expected to have different emission patterns in threes that have a considerable difference in age. Furthermore, the climatic variability among years makes it harder for comparison. The comparison is ok for the trees measured in 2014 so I would stick only to it.

Page 8 line 232: variability of what, please state. Page 8 line 234: if the tree number 2 has a different sampling technique than the other trees, can this be really comparable? Have you check the differences among sampling? Please make sure tree 2 and 3-8 are comparable to each other. Page 8 line 236: the values for monoterpenes were not statistically significant different from 0? Please state what you mean by significant. Page 8 line 242-244: please expand in how this study shows the importance of species specific measurements.

**3.4 Standard emission potential.**

As commented in the methodology, make a comparison between the temperature only and the temperature and light dependency, to see why the choosing of the algorithms makes sense.

Table 5: please change to bar graphs to see the comparison among species and seasons. Page 9 line 266: please insert similar behaviour to monoterpene emission potentials. Page 9 line 268-275: This section needs some reviewing in the sense that past studies have fit a temperature and light dependency emission dependency for carbonyl compounds (SHAO and Wildt, 2002). You mention that the best fit was obtained with the temperature dependent algorithm, please then state how better was as compared to the light and temperature dependency algorithm. Page 9 line 279: how this variability may reflect past temperature history or effects of incident or previous stress events? What is your explanation for saying this? Page 9 line 280: please state better what shall be taken into account, is past temperature history or effects of incident or previous stress events, or other? Page 9 line 281: what is reaction potential? Please
explain.

3.5 Total reactivity of emissions

You mention total reactivity of emissions, but you never give a total reactivity values, please do so, or else change to relative reactivity of emissions. Page 9 line 292: As you don't show these compounds in the graph, please state the contributions. Page 9 line 295: you mention Nölscher et al., 2013 paper, can you please state at what time of the year these measurements were carried out?

Conclusions

The first paragraph of the conclusion is just a brief summary of your results. The only actual conclusion I read is that the monoterpene emission pattern varies a lot (what is a lot?) from tree to tree. From your results and discussion I got the following messages, that if expressed as implications for boreal ecosystems can be used as conclusion from your study - What is the seasonality?  - There is low isoprene and moderate monoterpene emitters - Sqt emissions - Defence role b-farnense and linalool - OVOC roles, - Diurnal variability - Importance of tree to tree variability - Importance towards reactivity. Please redo the conclusions trying to show what are the take home message from your study.

References Guenther, A.B., Jiang, X., Heald, C.L., Sakulyanontvittaya, T., Duhl, T., Emmons, L.K., Wang, X., 2012. The Model of Emissions of Gases and Aerosols from Nature version 2.1 (MEGAN2.1): an extended and updated framework for modeling biogenic emissions. Geosci. Model Dev. Discuss. 5, 1503–1560. doi:10.5194/gmdd-5-1503-2012 SHAO, M., Wildt, J., 2002. Quantification of acetone emission from pine plants. Sci. China Ser. B 45, 532. doi:10.1360/02yb9070

---

## Referee Comment (RC2) · Anonymous Referee #1 · 2 Dec 2016

The study by Hakola et al. deals with emissions of Volatile Organic Compounds (VOC) from Norway spruce, a highly abundant tree species in boreal regions. The main focus has been given to monoterpenes (MT) and sesquiterpenes (SQT) while more species provide additional information on the emission strength and characteristics. The authors present 14 weeks of data that have been obtained over the course of three years but not at concise times. While such measurements are a valuable tool for any modeler, my main concern is that the study does not provide any new findings and the analysis performed is outdated and at many points confusing. It seems as a largely data description report and the editor should decide if such contribution fulfill the criteria of ACP and warrants a publication.

General comments:

[Figure]

- There is no concise conclusion. The authors state that the emissions were low in spring and early summer but increased during late summer and the maximums were located somewhere in July-August. I think that this is a rather abstract and un-quantitative conclusion.

- There are too less figures and to my opinion poor analysis. This makes the manuscript rather difficult to follow and drive conclusions.

- In the same context lies the fact that the authors chose to report results and discussion together. Since an overview on the existing studies is just discussed and not depicted in a table or figure, it's easy for the reader to get lost on the findings of other studies and deviate from the scope of the specific one.

- There is a mixture of trees, years and VOC species presented in a rather confusing way. I had to carefully note down all the details provided so I can follow the text which was not always easy. In addition, the different trees were of different age. I believe that greater attention shall be given in this "detail".

- The emission potentials. The authors derived the emission potential and the temperature dependency according to Guenther et al. (1993). Even if the core of current models is the same exponential algorithm, further improvements have been made. In addition, the R^2<0.1 is which is extremely low to be taken seriously. It would be very interesting to see how the all data lay on a graph together with temperature simulations. I'm afraid that it's dangerous from modeling point of view to report such strong temperature dependencies with such poor quality on the fit. You should at least discuss extensively.

I would have expected the authors to thoroughly analyze such an interesting dataset. I would therefore suggest major revisions addressing the greater picture. Is this temperature dependency and algorithm sufficient to describe the emissions from Norway spruce? How do current models compare with the measured emissions? What is the abundance of these species and how important are the emissions in case of extrap-

olation? Is the age of the tree important or we can assume similar emissions for all of them? Do you see any evidence of additional emission drivers apart from light and temperature? How important are Norway spruce emissions to the total reactivity of the boreal forests?

Specific comments:

L1. Acetone and acetaldehyde are barely reported to have a place in the title. Also "from Norway spruce" is misleading since the authors studied only trees in Finland. I would suggest to change the title into something more specific that would ideally include the main finding.

L18-L20. Please provide some standard deviation on the values reported. Emissions from conifers are usually reported per grams of dry weight as you did. However, I would appreciate an attempt to convert such emissions in area, if at all possible.

L24. The reported reactivity value lies on calculations and accounts for only the few measured VOC species. If it was measured, the authors would have probably seen the same contribution reported by (Nölscher et al., 2013). Since the SMEAR station implements a large suite of measurements for over a decade, I would suggest making a complete budget including inorganics before reporting that 70% of the OH reactivity comes from SQT. Please understand that such high value could be easily misinterpreted.

L48-L56. An important drawback of the study is the lack of clear objectives. Yes, we need more measurements and in situ GC-MS samples would be the ideal way of doing this. It is absolutely essential to evaluate temperature and light dependency but I have the feeling that this study does not go deep enough to assess these drivers in a boreal environment.

L61-63. You have measured five days in May 2011 and three (!) days in June. How can you be sure that from such short periods, you can derive a seasonal profile? Why

these days were characteristic for May, June and July respectively? Please provide some statistical evidence if this is the case. L65. What is the age of the 10meter tree?

L67. How many years younger than the 2011 tree? Can you provide evidence that a young tree behaves the same as an older one? Would that mean that if we plant some hectares of Norway spruce, in a couple of years their emission potential and general release of VOC would be similar to an old forest?

L71. Do you have evidence that PPFD strength is not changing by your enclosure? That would have large implications on the light driven VOC. Laboratory measurements assessing the absorbent strength of your enclosure are needed.

L72. Why did you choose to remove ozone at the inlet and not at the outlet? It has been shown that ozone can be a strong emission driver upon a given threshold. My objection here lies also on the fact that you are changing the conditions compared with the ambient.

L76-77. Allowing water vapor to your trap, will decrease the sensitivity of the MSD in a proportion similar to the ambient humidity during sampling. Were the calibrations performed also with wet air and at this trapping temperature? If not, your final values will be probably underestimated. Please provide a wet and dry calibration with the same setting and trapping temperatures to confirm that your approach was correct.

L88-89. I would suggest to completely remove acetone from the manuscript.

L96. Here is just an aforementioned comment that may make your manuscript more attractive to the modelling community: if it's possible, please convert the emissions to leaf area.

L104. Actually the parameterization in the models includes more variables, ecosystem characteristic. A detailed description can be found eg. in (Guenther et al., 2012). In general, I would suggest discussing over the current model algorithms assessing and evaluating all parameters.

L109. As you have shown in (Bourtsoukidis et al., 2012), environmental drivers such as high O3 abundance can also impact SQT emissions. Actually I'm a bit surprised to see that you have kept this study outside of your discussion.

L99-140. I don't see the reason why you have to repeat in text what is known since the last 23 years. I would recommend completely removing this part. Maybe you can replace it with a smaller one, but briefly discussing the current models.

L156. I strongly recommend to separate results and discussion.

L158-169. What is the reason of such presentation? I would suggest a plot or a less confusing approach that would directly allow the reader to distinguish the characteristics of each year.

L171. Please provide a number that indicates how much higher and how much significantly higher. Did you perform a p-test?

L193:198. The reasons for explaining the different seasonality are explained in a very broad way. It could also be the age of the tree, the pollution or simply the different climatic conditions.

L206. SQT may serve as signaling compounds as well eg. Vickers et al., 2009.

L230. In Fig. 1 you present a timeline. Diurnal variability would be better illustrated in a 24h plot and accounting for all days. Please include a figure where the diel cycle is presented for all the selected periods and years separately. Maybe then the reader can understand why you chose this period separation.

L245. The figure and the following results conclude otherwise. Please re-formulate the sentence. L233-L258. What is new when compared with Bäck et al. (2012)? I don't see any reason to include this tree variability in such detailed manner as it only confuses the reader and concludes on what is already known.

L277-278. Both StdErr and R2 indicate that a poor fitting for SQT during spring and

early summer. I would ask to include a figure with the SQT fittings, since this is the class of VOC you are mainly investigating. At which periods was the fitting best? At which worst? What can we learn from this? Even as supplement, this is more valuable than numbers which usually are taken for granted without further investigation on the other values provided.

L302-316. You actually present normalized contribution to OH reactivity from the species you measured. What is the reactivity of these emission measurements? How is it comparing with past measured values? From the values reported I would expect a small total reactivity that may be insignificant when compared with direct measurements. Including only the organics you measured and in the absence of a measured reactivity value, the result is kind of misleading. It creates the impression that SQT dominate the OH reactivity which is not the case. Or is it? Please calculate the reactivity including also the inorganic species measured at the station, report a value and compare with field measurements or from the literature. In general, I appreciate the effort to use OH reactivity, but the approach has to be slightly changed in order to address the bigger picture. I would be very impressed if SQT indeed dominate OH reactivity in a boreal environment.

L318-327. Your conclusions don't provide anything more than a description of the data. Please state what is the finding that makes your study suitable for publication.

Technical corrections:

There are still wrong abbreviations at lines 18,28,30,32,35,42,48,52,73,77,82,223. Maybe I've missed a couple of them but please understand that my brief report comment was suggesting a uniform terminology and not only the specific lines mentioned.

References:

Bäck, J., Aalto, J., Henriksson, M., Hakola, H., He, Q., and Boy, M.: Chemodiversity of a Scots pine stand and implications for terpene air concentrations, Biogeosciences, 9,

689-702, 10.5194/bg-9-689-2012, 2012.

Bourtsoukidis, E., Bonn, B., Dittmann, A., Hakola, H., Hellén, H., and Jacobi, S.: Ozone stress as a driving force of sesquiterpene emissions: a suggested parameterisation, Biogeosciences, 9, 4337-4352, 10.5194/bg-9-4337-2012, 2012.

Guenther, A. B., Jiang, X., Heald, C. L., Sakulyanontvittaya, T., Duhl, T., Emmons, L. K., and Wang, X.: The Model of Emissions of Gases and Aerosols from Nature version 2.1 (MEGAN2.1): an extended and updated framework for modeling biogenic emissions, Geosci. Model Dev., 5, 1471-1492, 10.5194/gmd-5-1471-2012, 2012.

Nölscher, A. C., Bourtsoukidis, E., Bonn, B., Kesselmeier, J., Lelieveld, J., and Williams, J.: Seasonal measurements of total OH reactivity emission rates from Norway spruce in 2011, Biogeosciences, 10, 4241-4257, 10.5194/bg-10-4241-2013, 2013.

Vickers, C. E., Gershenzon, J., Lerdau, M. T., and Loreto, F.: A unified mechanism of action for volatile isoprenoids in plant abiotic stress, Nat Chem Biol, 5, 283-291, 2009.

---

## Author Comment (AC1) · 17 Jan 2017

We wish to thank Referee 2 for valuable comments that improved our manuscript in many ways.

- There is no concise conclusion. The authors state that the emissions were low in spring and early summer but increased during late summer and the maximums were located somewhere in July-August. I think that this is a rather abstract and un-quantitative conclusion. The conclusions have been rewritten.

- There are too less figures and to my opinion poor analysis. This makes the manuscript rather difficult to follow and drive conclusions. We have added a new figure (Fig. 2 in revised MS) in to the main text and a graph describing the measurement system as supplement. . - In the same context lies the fact that the authors chose to report results

and discussion together. Since an overview on the existing studies is just discussed and not depicted in a table or figure, it's easy for the reader to get lost on the findings of other studies and deviate from the scope of the specific one. The chapter has been restructured - There is a mixture of trees, years and VOC species presented in a rather confusing way. I had to carefully note down all the details provided so I can follow the text which was not always easy. In addition, the different trees were of different age. I believe that greater attention shall be given in this "detail". We have clarified this and for example removed tree 2 from the chemodiversity study. - The emission potentials. The authors derived the emission potential and the temperature dependency according to Guenther et al. (1993). Even if the core of current models is the same exponential algorithm, further improvements have been made. In addition, the RËȨ2<0.1 is which is extremely low to be taken seriously. It would be very interesting to see how the all data lay on a graph together with temperature simulations. I'm afraid that it's dangerous from modeling point of view to report such strong temperature dependencies with such poor quality on the fit. You should at least discuss extensively. I would have expected the authors to thoroughly analyze such an interesting dataset. I would therefore suggest major revisions addressing the greater picture. Is this temperature dependency and algorithm sufficient to describe the emissions from Norway spruce? How do current models compare with the measured emissions? What is the abundance of these species and how important are the emissions in case of extrapolation? Is the age of the tree important or we can assume similar emissions for all of them? Do you see any evidence of additional emission drivers apart from light and temperature? How important are Norway spruce emissions to the total reactivity of the boreal forests? $R^2$ is an inadequate measure for estimating the goodness of nonlinear regression fits and it should not be used for this purpose (e.g. Spiess and Neumeyer 2010). However, many scientists and reviewers want it supplied with the nonlinear data analysis results, and this is why it is also given here. And all $R^2$ are not <0.1. Also, the measurements were carried in a natural forest environment, introducing many environmental factors which might affect the plants and their emissions. We have also found in earlier measurements that in Finland the temperature and light conditions are closely connected in summer, often leading to the saturation of the light algorithm, which limits the use of outdoor measurement results for testing or developing emission models (Hakola et al. 2006). In this work the fits were made for the whole data set, i.e. three years of measurement periods. This will affect the fits, because the conditions in different years and the response on the plants may vary a lot. If just testing different modeling approaches would have been the purpose of this exercise, it would have been better to carry out the measurements in a carefully controlled (laboratory) environment. Maybe also fitting all the outdoor measurement periods separately would have brought better correspondence, but this would have yielded several sets of emission potentials, serving no purpose for getting an average estimate of the emission behavior during the growing period. The measurements were classified as spring, early summer and late summer data groups, because this was the only way to characterize them during the season. Emission measurements and the model fits for some of the compounds are presented in Figures 1 - 7. The years are shown in separate panels, even though the analysis covered them together. From the results it can be seen, that the simple temperature controlled pool emission algorithm adequately covers all measurement periods, yielding the general levels of emission potentials for the spring, early summer and late summer classification. The emissions represent averages over all the years, so the observed emission strengths may be over or under predicted, and several emission peaks may be missed. But this is to be expected in this type of scattered measurement campaigns, when all conditions are not controlled or even measured, and where the plants are freely growing in their own natural environment. See also the discussion below, concerning the parameterization of the emission modeling.

Figure 1. $\alpha$-pinene emissions measured and predicted using the temperature dependent emission algorithm in the spring period.

Figure 2. Other sesquiterpenes emissions measured and predicted using the temperature dependent emission algorithm in the spring period.

Figure 3. $\alpha$-pinene emissions measured and predicted using the temperature dependent emission algorithm in the early summer period.

Figure 4. Other sesquiterpenes emissions measured and predicted using the temperature dependent emission algorithm in the early summer period.

Figure 5. $\alpha$-pinene and limonene emissions measured and predicted using the temperature dependent emission algorithm in the late summer period.

Figure 6. $\beta$-caryophyllene and other sesquiterpene emissions measured and predicted using the temperature dependent emission algorithm in the late summer period.

Figure 7. Acetone and aldehydes emissions measured and predicted using the temperature dependent emission algorithm in the late summer period.

Hakola H., Tarvainen V., Bäck J., Ranta H., Bonn B., Rinne J., and Kulmala M., 2006. Seasonal variation of mono- and sesquiterpene emission rates of Scots pine. Biogeosciences 3, 93-101. Spiess, A. and Neumeyer,N., 2010. An evaluation of R2 as an inadequate measure for nonlinear models in pharmacological and biochemical research: a Monte Carlo approach, BMC Pharmacology 10:6. doi:10.1186/1471-2210-10-6. Specific comments: L1. Acetone and acetaldehyde are barely reported to have a place in the title. Also "from Norway spruce" is misleading since the authors studied only trees in Finland. I would suggest to change the title into something more specific that would ideally include the main finding.

The MS has now a new name

L18-L20. Please provide some standard deviation on the values reported. Emissions from conifers are usually reported per grams of dry weight as you did. However, I would appreciate an attempt to convert such emissions in area, if at all possible. Standard deviations are included in the Table 3. A conversion factor from needle dry weight to needle area is now provided in chapter 2.2.

L24. The reported reactivity value lies on calculations and accounts for only the few

measured VOC species. If it was measured, the authors would have probably seen the same contribution reported by (Nölscher et al., 2013). Since the SMEAR station implements a large suite of measurements for over a decade, I would suggest making a complete budget including inorganics before reporting that 70% of the OH reactivity comes from SQT. Please understand that such high value could be easily misinterpreted. -Here calculated reactivity is the reactivity of the emissions and not ambient air reactivity. Therefore it is not possible to compare these with the compounds found in the ambient air. However, in summertime in ambient air at this site most of the known OH reactivity (which is ~50 % of total measured reactivity) is coming from the VOCs (Sinha et al. 2010). Other trace gases has lower contribution. In addition, aromatic hydrocarbons have only minor contribution compared to the terpenoids (Hakola et al. 2012). In those ambient air studies contribution of SQTs has been much lower than MTs, but those results are misleading, since lifetimes of the SQTs are so short that most of them are not detected in ambient air measurements and estimation of their contribution to the reactivity is possible only directly from the emissions. In the study of Nölscher et al (2013) measured also reactivity of the emissions and monoterpenes had major contribution to the total measured OH reactivity in the Norway Spruce emissions. However, they did most of their VOC measurements with PTR-MS, which is not the best methods to measure SQTs and we think that they could have missed major fraction of them. This is now clarified in the text in section 3.5 L48-L56. An important drawback of the study is the lack of clear objectives. Yes, we need more measurements and in situ GC-MS samples would be the ideal way of doing this. It is absolutely essential to evaluate temperature and light dependency but I have the feeling that this study does not go deep enough to assess these drivers in a boreal environment. We have added objectives for the study into the introduction. To assess drivers causing VOC emissions in boreal or any other vegetation area is a huge amount of work. We do not know what we are still missing and we do not know what causes seasonal variation and why it is so different in different places. In situ measurements can provide valuable new data to lead us few steps forward. Using gas-chromatograph has allowed us to determine

SQT emission rates and their seasonality together with aldehyde emission rates that has not been measured earlier. These affect greatly local atmospheric chemistry and they should be included in emission modelling. L61-63. You have measured five days in May 2011 and three (!) days in June. How can you be sure that from such short periods, you can derive a seasonal profile? Why these days were characteristic for May, June and July respectively? Please provide some statistical evidence if this is the case. L65. What is the age of the 10 meter tree? In 2011 we measured only 3 days in June, but in June 2014 two weeks. In 2011 in May measurements covered 5 days and in 2014 in May one week. More measurements would of course be useful, but we are quite confident that these measurements can describe the seasonal variability. Two years show similar results in terms of quantitative emissions although qualitatively monoterpene pattern varies.

The age of a 10 m tree is about 40 years. This has been added to the text.

L67. How many years younger than the 2011 tree? Can you provide evidence that a young tree behaves the same as an older one? Would that mean that if we plant some hectares of Norway spruce, in a couple of years their emission potential and general release of VOC would be similar to an old forest?

We definitely cannot provide evidence that the young trees behave the same as older ones. They seem to emit much less than big trees. We have highlighted this and concluded that the effect of age should be studied.

L71. Do you have evidence that PPFD strength is not changing by your enclosure? That would have large implications on the light driven VOC. Laboratory measurements assessing the absorbent strength of your enclosure are needed. Photosynthetically active radiation designates the spectral range of solar radiation from 400 to 700 nm. FEP film that is commonly used in reaction and emission chambers transmit solar radiation in the 290-800 nm region (see Finlayson-Pitts & Pitts: Chemistry of the upper and lower atmosphere).
L72. Why did you choose to remove ozone at the inlet and not at the outlet? It has been shown that ozone can be a strong emission driver upon a given threshold. My objection here lies also on the fact that you are changing the conditions compared with the ambient. This is true. We are changing the natural conditions. However, we were especially interested in sesquiterpene emissions and they are so reactive towards ozone that we would have missed a lot of them. Also, ozone scrubber cannot be placed in the outlet port because most of our compounds (all SQTs) would be lost there.

L76-77. Allowing water vapor to your trap, will decrease the sensitivity of the MSD in a proportion similar to the ambient humidity during sampling. Were the calibrations performed also with wet air and at this trapping temperature? If not, your final values will be probably underestimated. Please provide a wet and dry calibration with the same setting and trapping temperatures to confirm that your approach was correct.

We did not allow water to retain in the cold trap. The adsorbent material was hydrophobic and water passed the cold trap. To keep the cold trap dry we needed to keep the cold trap at 20 C temperature. This temperature was not cold enough to retain isoprene completely, so after 2011 we changed the trap material from Tenax-TA to dual trapping, Carbopack-B/Tenax TA. The trapping temperature was the same when analyzing emission and calibration samples.

L88-89. I would suggest to completely remove acetone from the manuscript. We decided to keep acetone in the manuscript. The calibration can be satisfactory although it is not linear. However we marked acetone as acetone/propanal,

L96. Here is just an aforementioned comment that may make your manuscript more attractive to the modelling community: if it's possible, please convert the emissions to leaf area.

We have measured leaf area of spruce needles at a site and weighted them. The conversion factor is added to the text in chapter 2.2.

L104. Actually the parameterization in the models includes more variables, ecosystem characteristic. A detailed description can be found eg. in (Guenther et al., 2012). In general, I would suggest discussing over the current model algorithms assessing and evaluating all parameters. The MEGAN model (Guenther et al. 2006; Sakulyanontvittaya et al. 2008; Guenther et al. 2012) for isoprene, monoterpene and sesquiterpene emissions has been developed with the goal of replacing regional emission inventories used to predict biogenic VOC emissions in the U.S.A. and globally. The model incorporates the leaf and branch-scale emission measurements, extrapolating them to canopy scale using a canopy environment model. The canopy model includes a leaf area index (LAI) which is estimated as 5, with 80% mature, 10% growing and 10% old foliage. The canopy is further divided into sun prone and shaded leaves which receive different solar radiation. The emissions are calculated based on plant functional types, and the process takes into account e.g. the canopy environment, the age of the leaves, and the soil moisture. The basic equations, are still the exponential temperature dependent mechanism and the light and temperature dependent formulation, where the light response is based on that of the photosynthesis, and the temperature term is based on the activity of isoprene synthase enzyme (Guenther et al. 1993). For monoterpene and sesquiterpenes emissions in MEGAN, Sakulyanontvittaya et al (2008) have described the temperature dependent emissions using the exponential equation. Additionally, they have assumed that 50% of sesquiterpenes and approximately 5-10% (with a few exceptions) of monoterpene species are emitted via the light and temperature dependent route. Guenther et al. 2006, 2012 also extend the light and temperature controlled emission to cover the average leaf temperature over the past 24 and 240 hours. Our measurements deal with fully sunlit branches, placed in Teflon enclosures for measuremtns in short periods during the growing season. Thus the modeling is carried out only to find any relation of the plant emissions with the direct emission processes. No modeling of sunlit or shaded leaves, effect of leaf age or temperature history, canopy environment, plant functional types and soil properties is carried out. No regional emission estimates that would benefit of a more broad approach are done. Modeling of the temperature

controlled pool emissions of monoterpenes and sesquiterpenes, and the light and temperature controlled isoprene emission are straightforward. In addition we also tested a hybrid algorithm which has both the temperature-dependent pool emissions and the instant light and temperature-dependent emissions combined. The hybrid algorithm did not produce more conclusive results when compared with the simple emission algorithms. Guenther A. B., Zimmerman P. R., Harley P. C., Monson R. K., and Fall R., 1993. Isoprene and monoterpene emission rate variability: Model evaluation and sensitivity analyses, Journal of Geophysical Research 98(D7), 12,609-12,627. Guenther A., Karl T., Harley P., Wiedinmyer C., Palmer P. I., and Geron C., 2006. Estimates of global terrestrial isoprene emissions using MEGAN (Model of Emissions of Gases and Aerosols from Nature), Atmospheric Chemistry and Physics 6, 3181-3210. Guenther A. B., Jiang X., Heald C. L., Sakulyanontvittaya T., Duhl T., Emmons L. K., and Wang X., 2012. The Model of Emissions of Gases and Aerosols from Nature version 2.1 (MEGAN2.1): an extended and updated framework for modeling biogenic emissions, Geosci. Model Dev., 5, 1471-1492, doi:10.5194/gmd-5-1471-2012. Sakulyanontvittaya T., Duhl T.,Wiedinmyer C., Helmig D., Matsunaga S., Potosnak M., Milford J., and Guenther A., 2008. Monoterpene and sesquiterpene emission estimates for the United States, Environ. Sci. Technol., 42, 1623–1629. L109. As you have shown in (Bourtsoukidis et al., 2012), environmental drivers such as high O3 abundance can also impact SQT emissions. Actually I'm a bit surprised to see that you have kept this study outside of your discussion.

In our set-up we had to remove ozone before the emission enclosure, therefore we were not able to study effects of ozone on emissions. However, 82 % of the measured O3 mixing ratios (N=21391) at the height of 4.2 m at SMEAR II in June-August 2015 were below the critical threshold (36.6 ppb) for correlation with ozone suggested by Bourtsoukidis et al. (2012). We have added this reference into the introduction.

L99-140. I don't see the reason why you have to repeat in text what is known since the last 23 years. I would recommend completely removing this part. Maybe you can

replace it with a smaller one, but briefly discussing the current models. This is a very good comment. The Emission potentials section has been rewritten, and only the key processes are named.

L156. I strongly recommend to separate results and discussion.

We have restructured the results and discussion to be clearer.

L158-169. What is the reason of such presentation? I would suggest a plot or a less confusing approach that would directly allow the reader to distinguish the characteristics of each year. The chapter has been rewritten.

L171. Please provide a number that indicates how much higher and how much significantly higher. Did you perform a p-test?

We have added box and whisker plots (Fig 1 in revised MS) to provide statistics of the measurements. L193:198. The reasons for explaining the different seasonality are explained in a very broad way. It could also be the age of the tree, the pollution or simply the different climatic conditions.

Unfortunately our data does not give any firm evidence what could cause the different seasonality. As you say, it can be age of the tree or climatic conditions. That is why more precise presentation is quite difficult. L206. SQT may serve as signaling compounds as well eg. Vickers et al., 2009. Vickers has been added to the text. L230. In Fig. 1 you present a timeline. Diurnal variability would be better illustrated in a 24h plot and accounting for all days. Please include a figure where the diel cycle is presented for all the selected periods and years separately. Maybe then the reader can understand why you chose this period separation. Figure 2 has been replaced by a new one as proposed by a reviewer

L245. The figure and the following results conclude otherwise. Please re-formulate the sentence. L233-L258. What is new when compared with Bäck et al. (2012)? I don't see any reason to include this tree variability in such detailed manner as it only confuses

the reader and concludes on what is already known. Bäck et al studied the chemotypes of Scots pine. Nobody has measured chemotypes of other tree species but Scots pine and therefore our finding that also Norway spruce has different chemotypes is a new important finding.

L277-278. Both StdErr and R2 indicate that a poor fitting for SQT during spring and early summer. I would ask to include a figure with the SQT fittings, since this is the class of VOC you are mainly investigating. At which periods was the fitting best? At which worst? What can we learn from this? Even as supplement, this is more valuable than numbers which usually are taken for granted without further investigation on the other values provided. See the above response to the comments on emission potentials, where also some observation & fitting plots are included. The nonlinear fitting should not be judged by R^2, because this is an unjustified measure for it (e.g. Spiess and Neumeyer, 2010). The sesquiterpenes emissions in spring were low, and the number of measurements was limited. Thus the spring results are only indicative. In early and late summer, the emissions were higher, and the simple temperature algorithm is able to predict the emission potential with much closer correspondence with the observations. Some of the peak emissions were not predicted, but the tested hybrid algorithm (which has both the pool emissions and the instant light dependent emissions combined) did not bring any closer results. Thus the reason for the emission peaks may be some other stimulus which the plant responds to, but which is not included in the simple modeling approach. Spiess, A. and Neumeyer,N., 2010. An evaluation of R2 as an inadequate measure for nonlinear models in pharmacological and biochemical research: a Monte Carlo approach, BMC Pharmacology 10:6. doi:10.1186/1471-2210-10-6.

L302-316. You actually present normalized contribution to OH reactivity from the species you measured. What is the reactivity of these emission measurements? How is it comparing with past measured values? From the values reported I would expect a small total reactivity that may be insignificant when compared with direct measurements. Including only the organics you measured and in the absence of a measured reactivity value, the result is kind of misleading. It creates the impression that SQT dominate the OH reactivity which is not the case. Or is it? Please calculate the reactivity including also the inorganic species measured at the station, report a value and compare with field measurements or from the literature. In general, I appreciate the effort to use OH reactivity, but the approach has to be slightly changed in order to address the bigger picture. I would be very impressed if SQT indeed dominate OH reactivity in a boreal environment. -it is not possible to include the inorganic species measured at the same site since these reactivities were calculated directly from emission measurements and not from ambient air data. Also comparing the values to the ambient air studies is not possible since the units are different. Therefore we decided to show relative values and title of the section was changed to clarify this. -On the other due to high reactivity of SQTs, most of them are not detected in ambient air measurements and it is possible to estimate their share to the local chemistry only directly from the emissions measurements. -At this site VOCs have higher contribution to the ambient air OH reactivity than other trace gases (NOx, CO, O3, CH4) especially in summer (Sinha et al. 2010). Monoterpenes are the main contributors to the total OH reactivity of the ambient air VOCs (Hakola et al. 2012) and based on the reactivities of the emissions, SQTs are actually more important than MTs to the local chemistry even though most of them are not detected in ambient air measurements due to the short lifetimes in air. This is now clarified in the text in section 3.5

L318-327. Your conclusions don't provide anything more than a description of the data. Please state what is the finding that makes your study suitable for publication.

We have rewritten the conclusions.

---

## Author Comment (AC2) · 17 Jan 2017

We wish to thank Referee 2 for many valuable comments. We have now corrected our manuscript accordingly.

Abstract Please state objectives and conclusions

They have been added

Introduction

It is missing an overview on the boreal forest. Norway spruce has only been mentioned once. Please rewrite the introduction taking into account a better explanation of the boreal ecosystem and the role that VOC emissions have in such ecosystems

We have written more about boreal forest and the BVOC emissions in the boreal

ecosystem.

What is the objective? Very few data on emissions? I suppose there is another ratio nale, please state.

More text about knowledge gaps in BVOC emissions from boreal area is added to the introduction.

Page 1, Line 14: please insert references that show forested boreal emissions of mt, sqt and OVOCs.

The references have been added.

Page 1, Line 16: contribute to the increase of methane lifetime? How? Please explain.

Oxidation of VOCs consume hydroxyl radicals and hence affect the lifetime of methane. This has been added to the text.

Page1, Line 26: You say in addition to isoprene, but is the first time you mention isoprene. Please expand.

We added isoprene also earlier in text, line 13.

Page 1, Line 28: please give an example of saturated aldehydes.

C4-C10 saturated aldehydes are given

Methods The methods sections needs considerable attention. The measurement times and tress are expressed in a confusing manner, and better explanation of sampling must be given.

We clarified the measurement protocol.

Another important issue is the comparison among trees. The tree measured in 2011 was different from the tree measured in 2014 and 2015, therefore I think they are not comparable as different processes such as age or different climatic conditions may come into play. This different tree should be removed from the comparison. Not necessarily from the study.

The tree measured in 2011 is not included in the chemodiversity study. Only measurements conducted during the same day are included. All the trees were different in chemodiversity study, because the idea was to show the diversity during the same day between individuals.

To begin with, a table with the different measured trees, years and techniques shall be stated.

Only two trees were measured (tree 1 in 2011 and tree 2in 2014 and 2015) and only one technique was used (in-situ gas-chromatographic measurements), so we do not think this needs a table. Additionally chemodiversity study was conducted during one day and then also trees 3-8 were measured, not with in-situ measurements but by taking adsorbent tube samples as shown in Fig 2.

We clarified this by adding more text.

Furthermore, a better explanation of the cuvette used is needed, a picture of the setting will help the reader considerably. Is there a blank cuvette? How do you take into the possible effects of the cuvette on the branch used?

We have added a picture of the set-up in supplementary material. There is no separate blank enclosure, but a blank can be measured by using empty enclosure. Branches can be harmed when they are enclosed in chambers and this can be seen in increased emissions. Therefore, we did not use the data until the emissions seemed settled. We also let the branch remain in the frame during the whole growing season, only the Teflon film was removed when the measurements were not conducted. This can be done without disturbing the plant.

Page 2, line 50: You say here the samples were collected. What do you mean by that?

We mean that the sample flow to the GC was directed from the branch located at about two meters height. The word 'collected' has now been changed to word 'taken'.

Which samples? GC cartridges? You take samples from the outlet of the cuvette?

We mean the sample flow to the GC from the enclosure outlet port. The sampling system was described in more detail in the text.

Please specify. Page 2 line 55: you mention you have a thermometer inside the enclosure. What brand? Is this thermometer having a possible artefact effect?

Thermometer conductor was covered with Teflon tubing and it is not supposed to cause any disturbance. The brand has been added to the text.

Page 2 line 56: how did you measure PPFD? Please include brand.

The brand has been added to the text.

Page 2 line 66: Please explain how the quantification of sabinene can be done using the calibration curve of b-pinene.

This is not an accurate method for quantification, but at least by using b-pinene calibration curve we can see how sabinene concentrations vary diurnally and seasonally. Sabinene, a-pinene and b-pinene have quite similar mass spectra and the ion 93 response of b-pinene is about 10 % larger than the response of a-pinene. Sabinene elutes very close to b-pinene in our system and therefore we used b-pinene response factor. Surely the error of sabinene measurements is higher. This has been added to the text.

Page 2 line 70: please include the manufacturer of the calibration solutions.

They have been added.

Page 3 line 72-75: here you say that in 2015 you were able to measure acetone and C4-C10 aldehydes. Then you say acetone was coeluted with propanal. Either you give a proof that you were able to properly calibrate acetone or you do not report acetone. In addition, please specify which C4-C10 aldehydes were you analysing.

The aldehydes measured are mentioned in the section 2.1. Calibration can be satisfactory although it is not linear.

Page 3 line 93: there are more recent studies (Guenther et al., 2012) that suggest the slope value should be 0.1.

Guenther et al. (2012) describes an update of the Model of Emissions of Gases and Aerosols from Nature (MEGAN) to version 2.1, which includes the emissions of approximately 150 specific compounds (classified into compound classes). MEGAN is a global model which is why the model parameters are set up to represent all biotopes and plant functional types in the terrestrial ecosystem. The model parameters have been developed based on the global database of Guenther et al. (1995), supplemented with results in several articles. The article cited for emissions in Europe is Karl et al. (2009), who consider a temperature dependent emission algorithm with slope value of 0.09 K-1 based on Guenther et al. (1993) for monoterpenes, and cite the results (0.17K-1) of Helmig et al. (2007) for sesquiterpene emissions. Section 2.3 Emission potentials has also been rewritten & made shorter. Guenther, A. B., Jiang, X., Heald, C. L., Sakulyanontvittaya, T., Duhl, T., Emmons, L. K., and Wang, X.: The Model of Emissions of Gases and Aerosols from Nature version 2.1 (MEGAN2.1): an extended and updated framework for modeling biogenic emissions, Geosci. Model Dev., 5, 1471-1492, doi:10.5194/gmd-5-1471-2012, 2012. Guenther, A. B., Hewitt, C. N., Erickson, D., Fall, R., Geron, C., Graedel, T., Harley, P., Klinger, L., Lerdau, M., McKay, W. A., Pierce, T., Scholes, B., Steinbrecher, R., Tallamraju, R., Taylor, J., and Zimmerman, P.: A global model of natural volatile organic compound emissions, J. Geophys. Res.-Atmos., 100, 8873–8892, 1995. Karl, M., Guenther, A., KÂĺoble, R., Leip, A., and Seufert, G.: A new European plant-specific emission inventory of biogenic volatile organic compounds for use in atmospheric transport models, Biogeosciences, 6, 1059–1087, doi:10.5194/bg-6-1059-2009, 2009. Guenther, A., Zimmerman, P. R., Harley, P. C., Monson, R. K., and Fall, R.: Isoprene and monoterpene emission rate variability: Model evaluations and sensitvity analyses, J. Geophys. Res., 98(D7), 12609–12617,

1993. Helmig, D., Ortega, J., Duhl, T., Tanner, D., Guenther, A., Harley, P., Wiedinmyer, C., Milford, J., and Sakulyanontvittaya, T.: Sesquiterpene emissions from pine trees – Identifications, emission rates and flux estimates for the contiguous United States, Environ. Sci. Technol., 41, 1545–1553, 2007.

Page 4 line 102: you say that you have used a temperature dependence for monoterpenes and a light and temperature dependence for isoprene. Please calculate also the temperature only dependence for isoprene and the light and temperature dependence for monoterpenes to conclude which is the best choice.

Several modeling approaches were tested on all compounds, including the traditional temperature only monoterpene-type pool emission dependence, the isoprene-type light and temperature instant emission dependence, and a hybrid algorithm with both pool and instant emissions. However, the results were not conclusive, and the temperature only relationship, which has also previously been found to correspond with the emission behavior of monoterpenes, covered the observed emissions well. For isoprene, the standard approach has generally been using the light and temperature dependent instant emission algorithm, and applying the other algorithms did not provide a better fit.

Page 5 line 128-133: This part is confusing. You need to properly explain how the sampling was performed in the different years. So for this day on the 24th of June of 2014, you analysed 6 different spruces which then you compare to the 7th tree which is the one continuously measured in 2014. I don't understand how can they be comparable if the sampling is different (tree number 7 uses the Teflon cuvette via the dynamic flow through, whereas the other 6 tress were sampled with a Teflon bag. Did you have a blank? For how long were you sampling? I also noticed that for the cuvette tree the adsorbents are different than for the Teflon bag, and disturbances can be different, therefore I would not compare them together. You need to give tree numbers from the beginning of the methodology, so it is clearer to the reader. Furthermore, a more detailed information about sampling and how this is different to the main sampling

is provided. I supposed these samples are analysed with the same instrument that is measuring cuvette air. Please state.

When taking samples only for qualitative purposes, as in this case, the sampling procedure is simpler. You do not need to know flow rates accurately and just few minutes sampling on tubes or on-line GC gives a monoterpene pattern that we were interested in. We have numbered the trees and described the sampling procedure better to make this clearer. The adsorbent in the tubes and in the GC cold trap was the same all the time. There is an error in the manuscript and this has now been corrected.

Page 5 line 139: you say that when experimental data was not available you use this software. Then use it to estimate the reaction coefficient for b-farnense and nitrate, as you mention its importance (linked to page 9 line 288).

There is no estimate available for nitrate+b-farnesene reaction

Results and discussion

3.1. Weather patterns during the measurements. Here you need a graph showing the year to year variability. In the table you cant really see what are the changes. Furthermore, in table 2 you say that those are mean values, therefore is needed to use standard deviations. For the rain you must state the mm, is it mm per month? A better explanation of how do you consider the seasons is needed, therefore perhaps pointing in the new meteorological figure when it is spring, early summer, late summer, etc, can help and then you express in the text why.

See response below. The section has been rewritten.

Page 5 line 145: you say temperatures are exceptionally high and precipitation is extremely low. what is high, what is low. This must be stated!

See response below. The section has been rewritten.

Page 5 line 151: you mention a warm spell in June and a cold spell in July. Please

show on new meteorological graph and explain what is a warm/cold spell. A clear and "based on meteorological data" election of the seasons must be stated. This can be added into the measurements table (i.e. year, tree, measurement technique (Cuvette, Teflon bag)

Section 3.1 Weather patterns during the measurements has been rewritten, with the purpose to only characterize the conditions in Finland during the growing season periods when the measurements were carried out, and say that they were in no way exceptional compared with the long-term averages. The weather patterns or meteorological data are not used to specify the seasonality for the measurements. The temporal distinction is only based on calendar months, spring months (March, April, May) and summer months (June, July, August). The summer months period was divided in early and late summer, because it has been observed in our earlier measurement campaigns that the emission speciation and emission rates may be different in early and late summer. (Tarvainen et al, 2005).

3.2. Variability of VOC emissions

Page 6 line 157: what do you mean by early season?

The word season is now replaced with the word summer, which is defined later in the sentence.

Page 6 line: 159-161: you cant compare the measurements of year 2011 to the measurements on 2014 and 2015 as they are different tree, so please only use 2014 and 2015 data for comparison. Furthermore, you present seasonal means and do no report of standard deviation. Please report standard deviations.

We are not comparing the trees, we want to give as representative value of the amount of compounds emitting to the atmosphere. Therefore, we think it is important to use all the data we have. The standard deviations are now included in the table.

Page 6 line 162: what is a low and a moderate emitter? Please report about values.

Furthermore, this kind of information suits much better in the introduction. It would also be good to have a comparison among other high emitting species from the boreal region.

We have added the emission rates we have cited. However, we decided to keep them here since it is easier to compare to our results. We do not really have high emitters in boreal region. Some birches emit monoterpenes in quite high amounts (Hakola et al., 2001), but seasonality of deciduous and coniferous trees is very different and comparison would not give very useful information.

Page 6 line 164: you say that you studies confirmed the low isoprene and moderate monoterpene emitters but the seasonal patterns were clearly different. Different to what? I have no indication of the seasonality of low isoprene and moderate monoterpene emitters, please mention and discuss.

The sentence "low isoprene and moderate monoterpene emitters" has been deleted" and the text concerning isoprene has been re-written.

Page 6 line166: this is the only mention of MBO (apart from the methodology). I would skip it or expand the explanation on MBO.

MBO sentence has been deleted.

Page 6 line 162-170: you report several emission rates but no uncertainty, please report.

We think that the text would be not nice to read if lots of numbers were included. Instead we wrote there Table 3, so readers were suggested to have a look at the Table and find the standard deviations there.

Page 6 line 171-172: you are still talking about monoterpenes and sesquiterpenes from your data, so this should go in the above paragraph.

The paragraphs have been combined

Table 3: In order to have an easier visibility of the data I prefer to see a bar graph of table 3, with uncertainties!

We have added standard deviations to the Table 3 as requested by the reviewer earlier and we think that numbers are more useful to most readers since then our figures can be compared with other results easier and they can be used by modellers. We have also added a new Figure 2 that describes the data statistics.

Page 6 line 171-184: here you do a comparison with Bourtsoukidis et al., 2014b. This is a nice comparison. But I prefer that first you mention the why of your found seasonality in the boreal forest to then start stating the difference to the German forests and thus differences in emissions. Furthermore, you mention only a difference between 0-84 ng g(dw)-1h-1 for SQT in your study, and this is a big part of your results. Please expand your SQT results and then compare to other studies.

The text has been restructured

Page 6 line 186: you say that the main sqt is b-farnense, can you comment about the other SQT measured?

$\beta$-caryophyllene and $\alpha$-humulene were also identified and this was added to the text. However, we observed several other SQT as mentioned, but since we did not have standards for them we cannot identify them conclusively. According to the mass spectra library there are usually many potential candidates for each of them and therefore we decided not to speculate what they could be.

Page 6 line 189: this is an important result and statement, therefore please show a graph showing the Linalool and sqt increase together. Furthermore this can be another conclusion from you study.

We have included a figure (Figure 2) showing monthly means of linalool, MT, SQT and aldehyde sum. Aldehydes and MT peak in July whereas SQT and linalool later in August. This was also added to the conclusions.

Page 7 line 193-207: This role should also be mentioned in the introduction. Furthermore, there has to be a better integration between the results from this study and the literature research.

The role has been added to the introduction.

Page 7 line 209: If you cant measure most volatile aldehydes then it does not make sense to say that the amount of measured carbonyl compounds was comparable to the monoterpenes, as it is misleading.

In this sentence we are not talking about aldehydes generally, but referring to the measured compounds. We do not think it is misleading. This just shows that also these emissions are significant.

Page 7 line 212: Could you provide with mean values for the percentages? Was this percentage calculated from both early and late summer, or they were calculated separately?

The percentages were for the whole summer. This has been added to the text. The mean values are shown in the Table and this has also been added to the text.

Page 7 line 213: you mention the possibility of bidirectional exchange when moist vegetation. Why? What is the link to your study? Please state.

The sentence has been deleted.

Figure 1: please include light as well to see the effect that light can have. Please remove/separate the graph from 2011 as it is not comparable to the other years as you were measuring a different tree. Please report as well standard deviations, name the compounds in the sum of C4-C10 aldehydes. If you were not able to give a proper explanation of the calibration for acetone, please remove from graph. In addition just a as help for the reader indicate which months comprehend the different selected seasons.

Light shows similar variation as temperature and we have shown in the modelling part

that has no effect on the SQT and MT emissions. It would not give any useful information and would make figures more unclear Early spring (April) measurements were only conducted in 2011 and in this Figure we want to show variability of all the compounds/compound groups that were the most meaningful each season. Therefore we would like to keep also April measurements in the Figure. Standard deviations cannot be reported since these are not averages. The names of the aldehydes have been added. We mentioned earlier that the calibration was satisfactory. It is not linear, but still it is satisfactory. The seasons are mentioned.

3.3. Tree to tree variability in emission pattern It is expected to have different emission patterns in threes that have a considerable difference in age. Furthermore, the climatic variability among years makes it harder for comparison. The comparison is ok for the trees measured in 2014 so I would stick only to it.

The tree 2 has been removed from the study

Page 8 line 232: variability of what, please state.

Variability in the monoterpene emission pattern. This has been added to the text

Page 8 line 234: if the tree number 2 has a different sampling technique than the other trees, can this be really comparable?

The tree 2 has been removed from the study

Have you check the differences among sampling? Please make sure tree 2 and 3-8 are comparable to each other.

See above

Page 8 line 236: the values for monoterpenes were not statistically significant different from 0? Please state what you mean by significant.

This has been added to the text

Page 8 line 242-244: please expand in how this study shows the importance of species specific measurements.

We have added more clarifying text.

3.4 Standard emission potential. As commented in the methodology, make a comparison between the temperature only and the temperature and light dependency, to see why the choosing of the algorithms makes sense.

Several modeling approaches were tested on all compounds, including the traditional temperature only monoterpene-type dependence, the isoprene-type light and temperature dependence, and a hybrid algorithm with both pool and instant light-dependent emissions. However, the results were not conclusive, and the temperature only relationship, which has also previously been found to correspond with the emission behavior of monoterpenes, covered the observed emissions well. For isoprene, the standard approach has generally been using the light and temperature dependent emission algorithm, and applying the other algorithms did not provide a better fit.

Table 5: please change to bar graphs to see the comparison among species and seasons.

We think that numbers are more useful to most readers since then our figures can be compared with other results easier and they can be used by modellers.

Page 9 line 266: please insert similar behaviour to monoterpene emission potentials.

Corrected

Page 9 line 268-275: This section needs some reviewing in the sense that past studies have fit a temperature and light dependency emission dependency for carbonyl compounds (SHAO and Wildt, 2002). You mention that the best fit was obtained with the temperature dependent algorithm, please then state how better was as compared to the light and temperature dependency algorithm.

Shao and Wildt refer to Guenther et al. (1993) and Tingey et al. (1991) for their algorithm for acetone emissions. Tingey et al. (1991) present a detailed monoterpene emission rate model which bases monoterpene emission rates on environmental conditions, leaf morphology, and needle resin content, with major emphasis on the effect of needle leaf temperature and the leaf structure. Guenther et al. (1993) discuss this and conclude that the Tingey et al. (1991) detailed model cannot be evaluated with existing field measurement data sets, and that detailed models require variables which are not available on regional scales. Guenther et al. (1993) present the temperature dependent one factor emission model M = MSexp($\beta$(T-TS)) for monoterpenes, and the multiplicative light factor (CL, where L is the photosynthetically active radiation PAR) and temperature factor (CT) controlled model I = ISCLCT for isoprene emissions. Equation (4) in Shuh and Wildt is not a combination of the models in Guenther et al (1993) and Tingey et al. (1991). It is a hybrid model utilizing the sum of the temperature controlled terpene-type pool emission factor and the temperature and light controlled isoprene biosynthesis-related emission factor in Guenther et al. (1993), with the modification of second power by Schuh et al. (1997). Furthermore, Guenther (1997) has corrected the formulation of the isoprene term to force the factor to be equal to 1 at standard light and temperature conditions (usually set at 30 °C and 1000 $\mu$mol photons m-2 s-1, Guenther et al. (1993); Kesselmeier and Staudt (1999); Wiedinmyer et al. (2004)), which is not included in equation (4) of Shao and Wildt (2002). Shao and Wildt (2002) measure pine plants under controlled environmental conditions in a continuously stirred tank glass reactor (CSTR), a 1600 L glass chamber. It is not explained where this reactor/laboratory is, or why Scots pine plants (Pinus sylvestris) are studied. Shao and Wildt (2002) measured acetone and isoprene emissions from pine for about half a year in spring-summer season (April to August). It is not clear how this seasonality is arranged for the measurements carried out in a chamber mounted in a temperature-controlled cell, with light provided by a set of Osram high intensity lamps. Also, it is not clear how the other measurement set covering April to December with a total of 7 measurements, described elsewhere in the article, was carried out.

In Figure 1 (emission rates of acetone and isoprene over the spring-summer period) caption the authors state that measurements were under leaf temperature of 25 °C and light intensity of 360 $\mu$E m-2 s-1. According to Wildt et al. (1997), who utilized a similar CSTR tank reactor and Osram lamps, their high light flux of 1090 $\mu$E m-2 s-1 corresponds to 30-40% of full sunlight. Isoprene emissions are not reported for Pinus sylvestris in Kesselmeier and Staudt (1999), a comprehensive overview of biogenic emissions, physiology and ecology. The trees Shao and Wildt (2002) observed, were very young, 2-3 years of age, and not growing in outdoor conditions, which means that their functionalities could be very different from trees growing in the field. Analysis results obtained in a controlled environment cannot be compared with field studies, where the environmental factors may pose conditions completely different to the laboratory surroundings. Also, the measurements we carried out in this manuscript are not involved with pines, we measured the emissions of adult Norway spruce (Picea abies), a different tree, in field environmental conditions. The emissions of different plants comprise different spectra of chemical compounds and there may be variations depending on the stresses or different environmental factors experienced by the plants. Our results yielded for spruce are only used to obtain indicative emission characteristics for the spring and summer period via simple fittings with the most common emission algorithms, not to compare any relative advantages or weaknesses of different emission processes.

Guenther, A., Zimmerman, P. R., Harley, P. C., Monson, R. K., and Fall, R.: Isoprene and monoterpene emission rate variability: Model evaluations and sensitvity analyses, J. Geophys. Res. 98(D7), 12609–12617, 1993. Guenther, A.: Seasonal and spatial variations in natural volatile organic compound emissions, Ecological applications 7(1), 34-45, 1997. Kesselmeier, J. and Staudt, M.: Biogenic volatile organic compounds (VOC): An overview on emission, physiology and ecology. J. Atmosph. Chem. 33, 23-88, 1999. Schuh, G., Heiden, A. C., Hoffmann, T., Kahl, J., Rockel, P., Rudolph, J., and Wildt, J.: Emissions of volatile organic compounds from sunflower and beech: Dependence on temperature and light intensity, Journal of Atmospheric

Chemistry 27, 291-318, 1997. Tingey, D. T., Turner, D. P., and Weber, J. A.: Factors controlling the emissions of monoterpenes and other volatile organic compounds. In: Trace Gas Emissions by Plants, Sharkey, T. D. et al., (eds), Academic Press, Inc. San Diego, California, 93-119, 1991. Wiedinmyer, C., Guenther, A., Harley, P., Hewitt, N., Geron, C., Artaxo, P., Steinbrecher, R., and Rasmussen, R.: Global organic emissions from vegetation. In: Emissions of atmospheric trace compounds, Granier, C. et al. (eds), Kluwer Academic Publishers, Dordrecht, 115-170, 2004. Wildt, J., Kley, D., Rockel, A., Rockel, P., and Segschneider, H. J.: Emission of NO from several higher plant species, J. Geophys. Res. 102(D5), 5919-5927, 1997.

Page 9 line 279: how this variability may reflect past temperature history or effects of incident or previous stress events? What is your explanation for saying this?

The past temperature history is a factor incorporated e.g. in the MEGAN modeling framework, similarly water stresses and other factors affecting plants are considered in the modeling work (e.g. Guenther et al. (2012)). In the text of the manuscript, the discussion on the various factors is only descriptive, we are not trying to guess in what ways these factors may affect the emissions.

Guenther, A. B., Jiang, X., Heald, C. L., Sakulyanontvittaya, T., Duhl, T., Emmons, L. K., and Wang, X.: The Model of Emissions of Gases and Aerosols from Nature version 2.1 (MEGAN2.1): an extended and updated framework for modeling biogenic emissions, Geosci. Model Dev., 5, 1471-1492, doi:10.5194/gmd-5-1471-2012, 2012.

Page 9 line 280: please state better what shall be taken into account, is past temperature history or effects of incident or previous stress events, or other?

This sentence has been deleted as it is not constructive.

Page 9 line 281: what is reaction potential? Please explain.

Reaction potential means the ability the compounds have to react

3.5 Total reactivity of emissions You mention total reactivity of emissions, but you never

give a total reactivity values, please do so, or else change to relative reactivity of emissions.

We have changed the title to relative reactivity

Page 9 line 292: As you don't show these compounds in the graph, please state the contributions.

Measured aldehydes do not react with O3 and therefore they are not found in Figure 4. However, they are shown in OH reactivity Figure and their average contribution is also mentioned in the text.

Page 9 line 295: you mention Nölscher et al., 2013 paper, can you please state at what time of the year these measurements were carried out?

This has been added to the text.

Conclusions The first paragraph of the conclusion is just a brief summary of your results. The only actual conclusion I read is that the monoterpene emission pattern varies a lot (what is a lot?) from tree to tree. From your results and discussion I got the following messages, that if expressed as implications for boreal ecosystems can be used as conclusion from your study - What is the seasonality? - There is low isoprene and moderate monoterpene emitters - Sqt emissions - Defence role b-farnense and linalool – OVOC roles, - Diurnal variability - Importance of tree to tree variability - Importance towards reactivity. Please redo the conclusions trying to show what are the take home message from your study.

More text has been added to conclusions

---

## Author Response (AR2)

Dear Prof. Aijun Ding

Please, find our corrections for the revised manuscript.

The authors have substantially improved their paper by reconstructing a large part of their initial submission and by providing a more comprehensive analysis. Even if they have ignored my suggestion for separating the results and discussion sections, the revised version reads well and justifies such decision. I'm also happy to see the presented model vs measurements comparisons but still curious to understand why b-caryophyllene and aldehydes do not fit the model. I would still suggest including these results even as supplement. However, this shall not be a prerequisite for publication. I believe that the current manuscript warrants a publication in ACP after addressing few minor issues.

*The modelling results are included in the Table 5 also for aldehydes and β-caryophyllene. Unfortunately, we cannot give the explanation why they do not fit better.*

L26. Please provide some % (including uncertainties) for a- and b-pinene.

*The % values with deviations have been added to the abstract.*

L92. I would suggest to include the setup in the main paper and not in the supplement.

*The setup has been placed to the main text.*

L105. The µm shall not be underlined.

*It is no longer underlined*

L184. "Were higher than normal". How much?

*The average temperatures in central Finland were 2-3 degrees higher than the normal long-term average temperatures. This has been added to the text.*

L185. "Had very little precipitation". How much?

*The precipitation was 70 % of the long term (30 years 1971-2000) average values in the whole country, and about 60 % in the central parts. This has been added to the text.*

L189. What is the long term average?

*Long term average means previous 30 years. This has been added.*

L366. It's not clear what you want to say. Please revise.

*The sentence has been revised and it is now* "It is also possible to measure total OH reactivity directly and experimental total OH reactivity measurements by Nölscher et al. (2013) showed that the contribution of SQTs in Norway spruce emissions in Hyytiälä was very small (~1%)."